# Sphingomyelin metabolism controls the shape and function of the Golgi cisternae

Felix Campelo[1]*, Josse van Galen[2,3], Gabriele Turacchio[4],
Seetharaman Parashuraman[4], Michael M Kozlov[5], María F García-Parajo[1,6],
Vivek Malhotra[2,6,3]*

[1]ICFO-Institut de Ciencies Fotoniques, The Barcelona Institute of Science and Technology, Barcelona, Spain; [2]Centre for Genomic Regulation, The Barcelona Institute of Science and Technology, Barcelona, Spain; [3]Universitat Pompeu Fabra, Barcelona, Spain; [4]Institute of Protein Biochemistry, National Research Council of Italy, Naples, Italy; [5]Department of Physiology and Pharmacology, Sackler Faculty of Medicine, Tel Aviv University, Tel Aviv, Israel; [6]Institució Catalana de Recerca i Estudis Avançats (ICREA), Barcelona, Spain

**Abstract** The flat Golgi cisterna is a highly conserved feature of eukaryotic cells, but how is this morphology achieved and is it related to its function in cargo sorting and export? A physical model of cisterna morphology led us to propose that sphingomyelin (SM) metabolism at the *trans*-Golgi membranes in mammalian cells essentially controls the structural features of a Golgi cisterna by regulating its association to curvature-generating proteins. An experimental test of this hypothesis revealed that affecting SM homeostasis converted flat cisternae into highly curled membranes with a concomitant dissociation of membrane curvature-generating proteins. These data lend support to our hypothesis that SM metabolism controls the structural organization of a Golgi cisterna. Together with our previously presented role of SM in controlling the location of proteins involved in glycosylation and vesicle formation, our data reveal the significance of SM metabolism in the structural organization and function of Golgi cisternae.

*For correspondence: felix.
campelo@icfo.eu (FC); vivek.
malhotra@crg.eu (VM)

## Introduction

The Golgi complex plays a central role in protein processing, sorting and transport (*Emr et al., 2009*). In higher eukaryotes the Golgi complex consists of multiple stacks of polarized flattened cisternae (*Klumperman, 2011*). Cisternae polarization allows for the directional transport and sequential processing of newly synthesized proteins arriving at the *cis*-face of the Golgi complex from the endoplasmic reticulum (*Glick and Luini, 2011*; *Stanley, 2011*). At the *trans*-Golgi network (TGN), fully processed proteins are sorted and exported to other compartments of the secretory pathway or for secretion (*Guo et al., 2014*). Remarkably, despite the large influx and efflux of membrane-bound transport carriers, the overall morphology of the Golgi complex remains essentially unaltered. How is the shape of the Golgi cisternae maintained and how does it relate to the function of this organelle? Golgi cisternae are characterized by having a relatively large area-to-volume ratio to accommodate the large numbers of incoming and outgoing transport carriers and also to efficiently regulate the enzymatic reactions occurring at the Golgi membranes (*Klumperman, 2011*). Moreover, a Golgi cisterna consists of two geometrically distinct regions with very different membrane curvatures: the central cisterna part, which is almost flat with the seldom presence of fenestrations or pores; and the highly bent rim of the cisterna. How the different functions of the Golgi membranes (namely, protein processing and transport) are organized between these two regions is not yet fully understood.

**eLife digest** The Golgi complex is a hub inside cells that transports many proteins to various parts of the cell. It also receives freshly made proteins and modifies them to help them mature into their final active forms. The complex is made up of a stack of disc-like membrane structures called cisternae.

Are the shapes of the cisternae important for the Golgi complex to work properly? Membranes are made of mixtures of molecules known as lipids and proteins. Previous experiments show that when the mixture of lipids in the Golgi membranes changes in a specific manner, the cisternae curl into an onion-like shape and the Golgi cannot process or send out proteins anymore.

Campelo et al. used mathematics and experimental approaches to investigate what causes the Golgi to change shape when the lipid mixture of the cisternae changes. A mathematical description of the shape of the Golgi predicted that some proteins keep the cisternae flat by holding the membrane rim that connects the two faces of a cisterna. To test this prediction, Campelo et al. performed experiments in human cells, which showed that when the mixture of lipids in the Golgi membranes changes, certain proteins jump from the rim, causing the cisternae to curl. These same proteins are also needed to transport cargo proteins out of the Golgi, meaning that there is a connection between the shape of the Golgi and the tasks it carries out.

The shape of the Golgi complex is altered in Alzheimer's disease and many other neurodegenerative diseases. The next challenges are to understand how these shape changes happen, how this affects cells, and if it could be possible to develop drugs that prevent these changes from occurring in patients.

We previously reported that disruption of sphingomyelin (SM) organization specifically at the Golgi membranes –by SM synthase-mediated formation of short-chain SM at the *trans*-Golgi membranes– leads to inhibition of transport carrier formation (*Duran et al., 2012*) and also to defects in transmembrane protein glycosylation and localization (*van Galen et al., 2014*). Interestingly, we showed that these effects occur concomitantly with an overall reduction in the lateral lipid order of the Golgi membranes (*Duran et al., 2012*) as well as with striking alterations in the morphology of Golgi cisternae, which abandon their typical flat morphology and become highly curled (*van Galen et al., 2014*). Based on our findings, we suggested that short-chain SM might not be able to generate liquid-ordered nanodomains at the Golgi membranes (*Duran et al., 2012*). However, it is still unclear whether there is any causal relation between the ability of SM to control lateral Golgi membrane organization and the observed changes in the morphology of the Golgi cisternae.

Motivated by these experimental observations, we decided to investigate the physical mechanisms by which SM metabolism controls Golgi cisternae morphology, with a general aim at understanding whether the shape and the function of the Golgi complex are two sides of the same coin and how they relate to each other. Curling of a flat Golgi cisterna has, from a physical point of view, severe consequences. A flat cisterna has a large surface area at its rim with a very high membrane curvature thereby bearing large elastic stresses (*Shibata et al., 2009*). Hence, cisterna curling is accompanied by a change in the distribution of the membrane elastic stresses. The quantitative analysis of the extent of these stresses and how they can be sustained within the overall cisterna morphology requires a physical description of the Golgi cisterna free energy. Here we present a biophysical model that describes the free energy of a Golgi cisterna to elucidate the mechanisms by which SM homeostasis mechanically regulates the shape of the Golgi complex and therefore its function. In the following, we describe the model, the results derived from it and the experimental validation of the model's predictions.

## Results

### Theoretical results

### Qualitative description of the proposed mechanisms of SM-regulated Golgi morphology

The shape of cellular organelles, such as the Golgi complex, results from the generation and stabilization of the curvature of their membranes (*Shibata et al., 2009*). According to the elastic model of membrane bending (*Helfrich, 1973*; *Campelo et al., 2014*), energy is required to induce local changes of the membrane curvature from its preferred or spontaneous curvature. This energy is generally supplied by specialized lipids and/or membrane proteins, usually referred to as membrane curvature generators (*Zimmerberg and Kozlov, 2006*; *McMahon and Gallop, 2005*; *Kozlov et al., 2014*). Moreover, the amount of bending energy associated with local curvature deviations is proportional to the local bending rigidity of the membrane (*Helfrich, 1973*). As a consequence, both local variations in the amounts of curvature generators present on the membrane and in the bending rigidity of the membrane can influence the morphology of Golgi cisternae. Taking into account these considerations, our aim here is to establish a physical model for the Golgi membrane morphology, with a special focus on understanding the mechanisms by which SM metabolism controls the overall shape of the Golgi cisternae. Similar models based on the Helfrich bending energy have been widely used in the past to describe the shapes of lipid vesicles (*Seifert, 1997*). In such models, transitions from flat cisterna-like vesicles to curled vesicles, named stomatocytes, were promoted upon a reduction in the volume-to-area ratio. We previously proposed that reduction in the volume-to-area ratio of the Golgi cisternae could in fact be responsible for the observed flat-to-curled cisternae transition upon short-chain ceramide treatment (*Duran et al., 2012*). However, in our subsequent studies, we observed no obvious increase in the overall volume-to-area ratio of the Golgi cisternae during such a morphological transition (*van Galen et al., 2014*). Moreover, Golgi cisternae have an extremely low volume-to-surface ratio, which does not account for the reported flat configurations according to the aforementioned models (*Seifert et al., 1991*; *Miao et al., 1991*; *Seifert, 1997*). Altogether, this prompted us to propose an alternative model that takes into account two contributions that could potentially influence a role in SM-regulated shaping of the Golgi cisternae: (i) the presence of small, rigid, and highly dynamic membrane nanodomains enriched in sphingolipids and cholesterol; and (ii) the SM-dependent recruitment to or release from the Golgi membranes of budding factors and other membrane curvature generators essential for the formation of transport carriers. We first qualitatively describe how each of these two contributions can influence the shape of a Golgi cisterna.

### Nanodomain partitioning-mediated mechanism

Experimental evidence has suggested the existence of nanoscopic SM-enriched liquid-ordered domains at the Golgi membranes, although direct visualization has remained challenging (*Gkantiragas et al., 2001*; *Klemm et al., 2009*; *Duran et al., 2012*; *Bankaitis et al., 2012*; *Deng et al., 2016*). Based on in vitro data, such liquid-ordered membrane domains are expected to have a higher bending rigidity than the surrounding liquid-disordered membrane and are therefore less prone to accommodate membrane curvature (*Roux et al., 2005*; *Heinrich et al., 2010*; *Dimova, 2014*). In essence, the presence of large amounts of such rigid nanodomains at the highly curved rim of a flat Golgi cisterna is associated with a large bending energy penalty. There are two ways to reduce this bending energy. The first one is by partitioning these rigid nanodomains away from the rim to the flatter regions of the Golgi cisterna (*Figure 1A*). However, such inhomogeneous curvature-driven nanodomain redistribution is entropically unfavorable. Therefore the balance between the bending energy and the entropic free energy dictates the optimal distribution of nanodomains along the cisterna membrane (*Figure 1A*). The second possibility is to decrease the surface area of the highly curved rim, hence reducing the overall bending stress of the rim, by globally changing the shape of the Golgi cisterna from a flat to a curled configuration, while maintaining the total surface area of the cisterna (*Figure 1A*). This morphological transition is associated with a reduction of the bending energy of the rim, but also with an increase in the bending energy of the central region of the cisternae (*Knorr et al., 2012*; *Helfrich, 1974*). Again, the balance between these two opposite contributions to the overall bending energy determines the optimal cisternae shape (*Figure 1A*). Obviously, these two means of decreasing the cisternae free energy are not

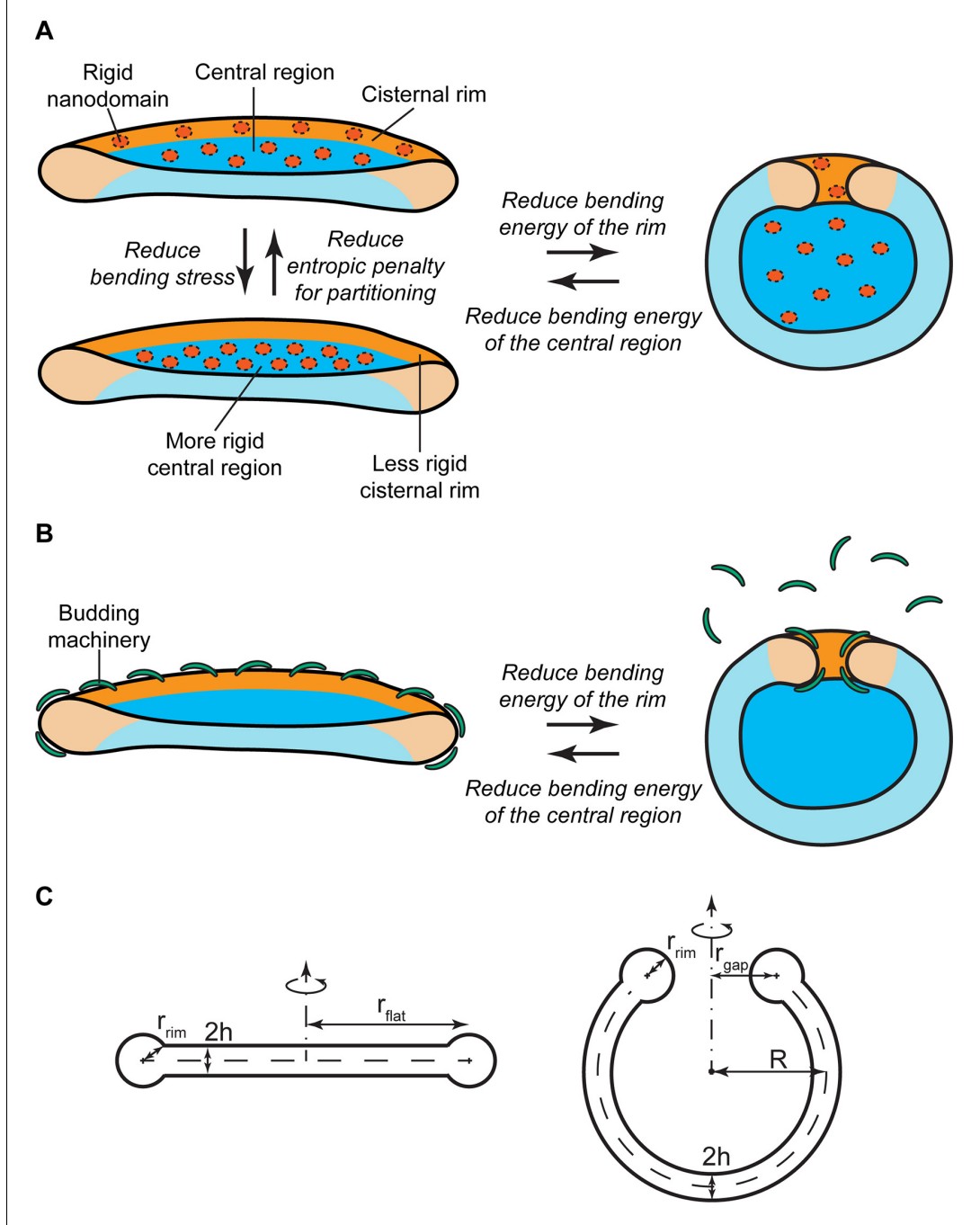

**Figure 1.** Mechanisms of SM-regulated Golgi cisternae morphology. (**A**) Nanodomain partitioning-mediated mechanism. A schematic representation of flat (left) and curled (right) Golgi cisternae is shown, where the highly bent cisternae rims and the flatter central regions are colored in orange and blue, respectively, and rigid nanodomains in red. Non-homogeneous partitioning of rigid nanodomains from the rim to the central region of a flat cisterna (left) reduces the overall bending stress but increases the entropic free energy penalty of partitioning. Flat-to-curled cisterna transition (left to right) reduces the bending energy of the rim in expenses of an increase in the bending energy of the central region. (**B**) SM-dependent release of budding effectors. Membrane curvature generating proteins, such as components of the budding machinery, present at the cisterna rims are shown in green. Partial release of these proteins from the membranes can lead to destabilization of the flat cisterna rim, and a flat-to-curled cisterna transition can be triggered depending on the balance between the bending energies of the rim and flat regions. (**C**) Cross section of the geometry of the flat (left) and curled (right) cisterna configurations used in our model. The vertical axes are the axes of symmetry.

The following figure supplement is available for figure 1:

*Figure 1 continued*

**Figure supplement 1.** Geometry of the cisterna rim.

mutually exclusive. Instead, a combination of both lateral nanodomain partitioning and a change in Golgi cisternal shape might possibly result from a decrease in the amounts of SM-enriched nanodomains present at the Golgi membranes (*Figure 1A*).

## SM-dependent release of budding effectors

The budding machinery, such as the COPI and clathrin coats, is responsible for generation of large membrane curvatures that are required to form transport carriers at the Golgi membranes (*Kirchhausen, 2000*; *Campelo and Malhotra, 2012*). It is reported that budding events are more frequent at the rims of the Golgi cisternae, where the membrane is already curved thus facilitating this process (*Rothman, 2010*; *Lavieu et al., 2013*; *Popoff et al., 2011*; *Engel et al., 2015*). In addition to their role in transport carrier formation, recruitment of components of the budding machinery to the cisterna rims helps stabilizing these highly bent membranes (*Figure 1B*). Hence, a reduction in the amounts of membrane curvature generators present at the cisterna rim leads to an increase of the bending energy penalty of the rim, which could in turn be relaxed by a flat-to-curled shape transition (*Figure 1B*).

## Prediction of flat and curled cisterna configurations

In order to quantify the relative effect in Golgi shaping of the two mechanisms proposed above, we developed a mathematical formulation of the free energy of a Golgi cisterna as a function of its shape and nanodomain distribution on the membrane. The major assumption of our model is that the time scale of mechanical equilibration of the cisterna shape is much smaller than that of the changes in lipid and protein composition through membrane fluxes. Non-equilibrium shapes should be considered only if the composition would change faster than the shape relaxed to the new equilibrium state (*Sens and Rao, 2013*). We can estimate the mechanical relaxation time as a combination of the characteristic viscosity, bending rigidity and length scale of the cisterna, $\tau_{mech} = \eta R^3/\kappa \sim 1\ ms$ (*Allain et al., 2004*). On the other hand, the rates of the composition changes based on the fluxes through the Golgi cisternae have been theoretically inferred from experimental data (*Dmitrieff et al., 2013*), resulting in a characteristic compositional relaxation time by means of membrane fluxes, $\tau_{flux} \sim 100\ s$. Since $\tau_{mech} \ll \tau_{flux}$, the cisternae shape is assumed to be mechanically equilibrated for every instant composition. Since the composition is in a steady state, the shape is in mechanical equilibrium corresponding to this steady state composition. Hence, the equilibrium configuration of a cisterna is assumed to correspond to a free energy minimum.

We consider the cisterna membrane to have a shape of a sheet bound by a rim (*Figure 1C*). The sheet part is represented by two parallel membranes with inter-membrane distance, $2h$, much smaller than the sheet lateral dimension, $r_{flat}$. The maintenance of the narrow luminal space of Golgi cisternae can result from protein arrays bridging the two parallel membranes of a cisterna, which have been visualized by cryo-electron tomography (*Engel et al., 2015*). Alternatively, membrane adhesion between adjacent cisternae has been shown to be required to keep the narrow luminal space in HeLa cells (*Lee et al., 2014*). The rim shape is modeled by an open toroid of a cross-sectional radius, $r_{rim}$, merging the sheet boundary (*Figure 1C*). The distance between the bridging and/or stacking protein scaffolds forming the flat part of a cisterna and the cisterna edge sets the cross-sectional radius of the cisterna rim. The sheet part of the cisterna can curve into spherical segments of variable radii, $R$, which is accompanied by the corresponding changes of the rim perimeter, $L = 2\pi r_{gap}$, the latter measured as the length of the toroidal axis. We will use the shape parameter $r_{gap}$ as a means to quantitate the degree of cisternal curling.

The free energy responsible for the cisterna shape includes the elastic bending energy of the membrane sheet and rim and also the entropic cost of a non-homogeneous partitioning of rigid nanodomains between the sheet and the rim. The bending energy is computed based on the Helfrich model (*Helfrich, 1973*), in which the membrane elastic properties are characterized by the bending modulus, $\kappa$, and the spontaneous curvature, $J_s$, the latter describing the intrinsic tendency

of the membrane to curve (see the Materials and methods for a complete description of the model). We assume that the membrane spontaneous curvature is generated by specific proteins or proteins complexes (such as the components of the budding machinery) bound to or inserted in the outer membrane monolayer. These proteins occupy a fraction $\phi_{budding}$ of the outer monolayer area, and are characterized by an effective individual spontaneous curvature, $\zeta_{budding}$, which has typical values in the range $\zeta_{budding} \approx 0.5 - 0.75\ nm^{-1}$ (*Campelo et al., 2008*). The membrane spontaneous curvature is given by $J_s = \frac{1}{2}\phi_{budding}\zeta_{budding}$ (the factor ½ accounting for the resistance of the internal monolayer to curving of the external one) and can vary along the membrane in accord with variation of the area fraction, $\phi_{budding}$. (*Campelo et al., 2008*).

The membrane nanodomains are considered to occupy a fraction Φ of the overall membrane area. The nanodomains are assumed to have a vanishing spontaneous curvature and a bending rigidity greatly exceeding that of the surrounding membrane (*Roux et al., 2005*; *Heinrich et al., 2010*). The domains can freely partition between the rim and the sheet parts of the system. The detailed presentation of the system free energy is given in the Materials and methods section. In essence, the relative contribution between the free energy of the rim and that of the rest of the cisterna mainly governs the transitions between flat and curled shapes, in analogy to other membrane systems (*Helfrich, 1974*; *Lipowsky, 1992*; *Knorr et al., 2012*).

We determined the equilibrium shape of a Golgi cisterna by minimizing the free energy for a given set of geometric and elastic parameters (see *Table 1*) upon specific assumptions. First, we consider the budding machinery to localize exclusively at the rims of the Golgi cisternae, so that the membrane of the sheet part the cisterna has a vanishing spontaneous curvature $J_{s,mid} = 0$ while the rim membrane is characterized by $J_{s,rim} = J_s$ (see Materials and methods). Second, we assumed that the area fraction of the curvature generators in the rim, $\phi_{budding}$, ranges between 0% and 10%, so that the variation range of the membrane spontaneous curvature in the rim is $0 \leq J_s \leq 0.033\ nm^{-1}$. And third, the overall membrane area fraction covered by the nanodomains, Φ, varies over a wide range of values, $0 \leq \Phi \leq 0.4$.

Our model predicts that, depending on the values of the two parameters, $J_s$ and Φ, the minimal energy state of the system can correspond to a flat cisterna or a highly curled cisterna. In an alternative situation, referred below as the bistability state and considered in more detail in the next section, both the flat and curled shapes correspond to local energy minima. The parameter ranges corresponding to the three possible states of the system are summarized in a shape diagram (*Figure 2A*). Qualitatively, the model predicts that increasing the spontaneous curvature of the rim membrane by augmenting the amount of the curvature generators in the rim favors a shape transition from the curled to the flat cisterna configuration. In other words, large numbers of the budding factors at a Golgi cisterna favor flat rather than curled cisternae (see *Figure 1B*). Moreover, according to our computations, transitions between curled and flat cisternae are almost insensitive to variations of the nanodomain area fraction, Φ (*Figure 2A*). The reason being that the free energy required to partition large amounts of rigid nanodomains away from the curved rim cannot be counterbalanced by the relaxation of the bending energy (see Appendix).

**Table 1.** Model parameters.

| Parameter | Value(s) |
| --- | --- |
| $r_{flat}$ | 500 nm; 1000 nm |
| $r_{rim}$ | 30 nm |
| $h$ | 15 nm |
| $R_d$ | 5 nm; 2 nm; 20 nm |
| Φ | 0–0.4 |
| $\kappa_{ld}$ | 20 $k_B$T |
| $\kappa_{lo}$ | 80 $k_B$T |
| $\alpha_{\bar{\kappa}}$ | −0.83 |
| $J_s$ | 0–0.033 $nm^{-1}$ |

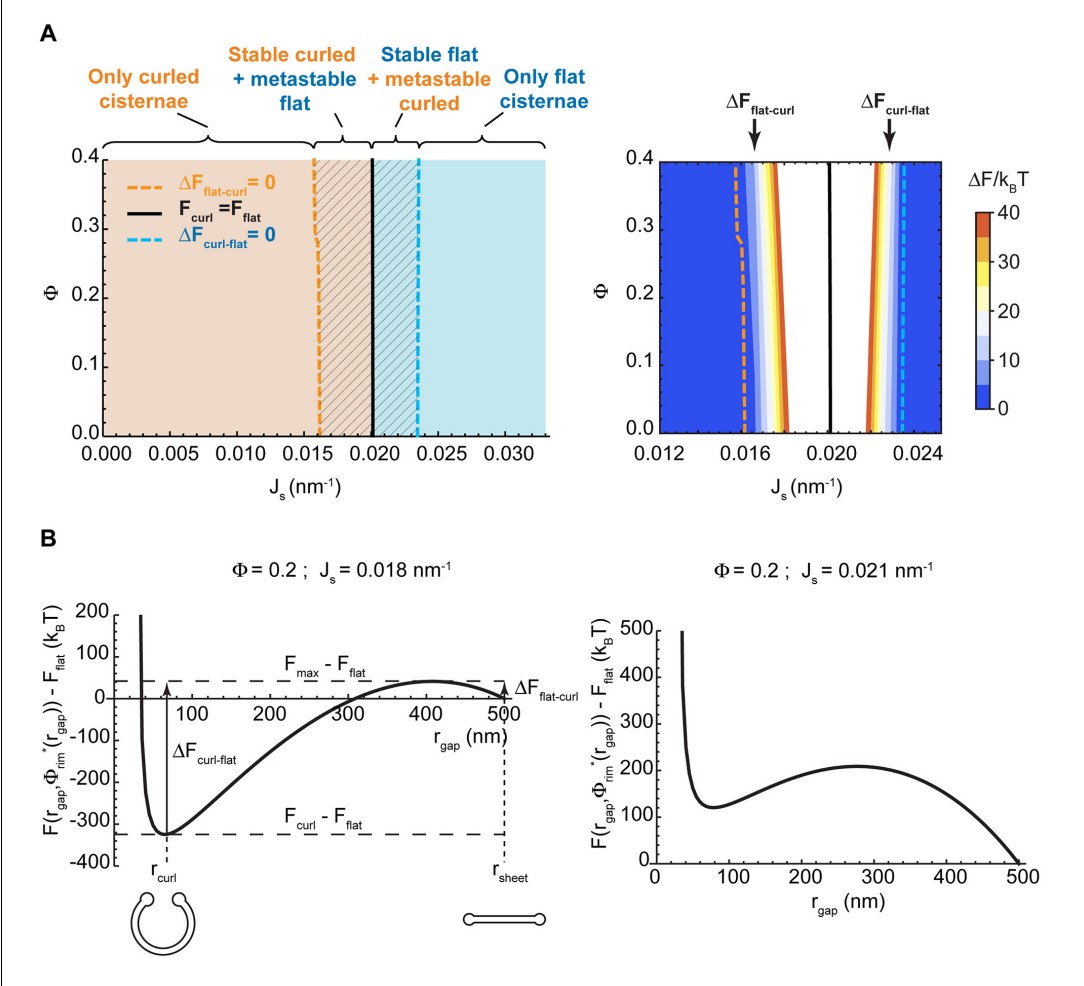

**Figure 2.** Shape diagram of a Golgi cisterna as a function of the area fraction of nanodomains and the membrane spontaneous curvature. (**A**) The existence of locally stable flat and/or curled cisternae was quantitatively assessed using our model of cisterna morphology, and the results depicted in a shape diagram for different values of the area fraction covered by nanodomains, $\Phi$, and of the membrane spontaneous curvature, $J_s$, (left panel). Four regions can be distinguished: a region where curled cisternae are the only locally stable shapes (orange), a region with only flat cisternae (blue), and two regions where curled and flat configurations are respectively stable and metastable (orange dashed) or metastable and stable (blue dashed). The designated orange, black and blue lines indicate the boundaries between these regions. In the right subpanel, the energy barriers for the flat-to-curled or curled-to-flat transitions within the bistability regions are shown and color-coded (only energy barriers smaller than $40\ k_BT$ are shown for clarity). (**B**) The total free energy of a cisterna with respect to the flat configuration as a function of the degree of curling, $r_{gap}$, for two different sets of parameters.

The following figure supplements are available for figure 2:

**Figure supplement 1.** Effect of the distribution of budding effectors along the cisterna on the shape diagram.

**Figure supplement 2.** Effect of the nanodomain area fraction on the shape transitions.

Additionally, we considered the situation where the budding machinery is homogeneously distributed along the whole Golgi cisterna, $J_{s,rim} = J_{s,mid} = J_s$, and compared the model predictions with the previous case of the curvature generators concentrated only at the rim (**Figure 2A**). Since the flat-to-curled cisterna transition is associated with an increase in the surface area of the central part of the cisterna, the presence of curvature generators therein leads to an increase in the total bending energy of this area, thereby opposing cisterna curling. Indeed, our results show no overall qualitative difference from the results in **Figure 2A**, but only a general shift of the shape transitions between the flat and curled cisterna configurations towards the lower values of the membrane spontaneous

curvature, as well as a reduction of the parameter space occupied by the transition area (*Figure 2—figure supplement 1*). In the Appendix, we present an analytical estimation of the extent of this shape transition shift.

In summary, the model predicts that the presence of membrane curvature generators at the Golgi cisterna is necessary to stabilize a flat morphology and a partial release of such curvature generators destabilizes the flat shape in favor of a curled cisterna shape. In the next sections we expand, describe and analyze these results in more detail.

## State of the system bistability

The bistability region of the shape diagram (*Figure 2A*, left, shaded region) encompasses a set of values of the spontaneous curvature of the cisterna rim, $J_s$, and of the nanodomain area fraction, $\Phi$, for which the free energy has two local minima. Each of these two local minima corresponds to a locally stable cisterna configuration. This is illustrated in *Figure 2B*, which represents the total free energy of a Golgi cisterna as a function of the shape parameter, $r_{gap}$, for two sets of the parameter values within the bistability region (see *Figure 2A*). The two local minima of the energy correspond, respectively, to a highly curled and a flat cisterna (*Figure 2B*, left panel). Moreover, an energy barrier separates the two locally stable shapes. Hence, at any transition from a curled to a flat cisterna shape the system needs to overcome an energy barrier $\Delta F_{curl-flat} = F_{max} - F_{curl}$; whereas transition from the flat to the curled morphology requires crossing an energy barrier $\Delta F_{flat-curl} = F_{max} - F_{flat}$ (see *Figure 2B*). We computed the values of these energy barriers for both flat-to-curled and curled-to-flat transitions, which are shown in *Figure 2A* (right panel). Our results predict a relatively broad range of parameters within the bistability region of the shape diagram where the shape transition can occur by crossing relatively low energy barriers, comparable to the typical few $k_B T$ energies of thermal fluctuations (see the color-coded plot in *Figure 2A*, right panel).

## Control of the flat-to-curled cisterna transition by the membrane curvature generators

Our model predicts that the Golgi cisterna shape transition must be driven, most effectively, by variations in the amount of curvature generators present on the membranes of the Golgi rims (*Figure 2A*). To explore this mechanism in more depth, we considered a situation where the rim spontaneous curvature, $J_s$, is allowed to vary within a range, $0 \leq J_s \leq 0.033 \ nm^{-1}$, whereas the membrane area fraction covered by nanodomains, $\Phi$, is taken to be constant and equal to 0.2. Then, for each value of $J_s$, we compute (see Materials and methods) (i) which of the two possible cisterna shapes corresponds to the free energy minimum, that is, the equilibrium state; (ii) the energy value in the equilibrium state determined with respect to the flat cisterna configuration, and (iii) for the bistability range, the energy barrier between the two minimal energy states. The results, shown in *Figure 3A*, indicate that for large values of the spontaneous curvature, $0.024 \ nm^{-1} < J_s \leq 0.033 \ nm^{-1}$, the only possible stable cisterna configuration is the flat one (solid blue line, *Figure 3A*). An intermediate range of spontaneous curvatures, $0.016 \ nm^{-1} < J_s < 0.024 \ nm^{-1}$, corresponds to the bistability state where both curled and flat Golgi cisternae (orange and blue lines, respectively, *Figure 3A*) are locally stable shapes separated by a free energy barrier. Finally, for lower values of the spontaneous curvatures, $0 \leq J_s < 0.016 \ nm^{-1}$, the only stable shape is that of a curled Golgi cisterna (solid orange line, *Figure 3A*).

Next we focused on the shape bistability region, $0.016 \ nm^{-1} < J_s < 0.024 \ nm^{-1}$, and computed the energy at the peak of the energy barrier between the two shapes, $F_{max}$ (*Figure 3A*, black solid line). This allowed us to compute the energy barrier required to be overcome for transition from a curled to a flat cisterna, $\Delta F_{curl-flat}$; and the energy barrier of the inverse transition from the flat to the curled, $\Delta F_{flat-curl}$ (*Figure 3B*). In addition, we computed the relative redistribution of nanodomains between the rim to the sheet part of the cisterna for both flat and curled cisternae shapes. These results show that there is a preferential partitioning of nanodomains from the highly curved rim region to the flatter sheet region of the cisterna, and that this partitioning is enhanced upon a reduction in the spontaneous curvature of the Golgi cisterna (*Figure 3C*). Notably, these results further predict the non-homogeneous distribution of nanodomains along the Golgi membranes to be more pronounced in curled as compared to flat cisternae (*Figure 3C*).

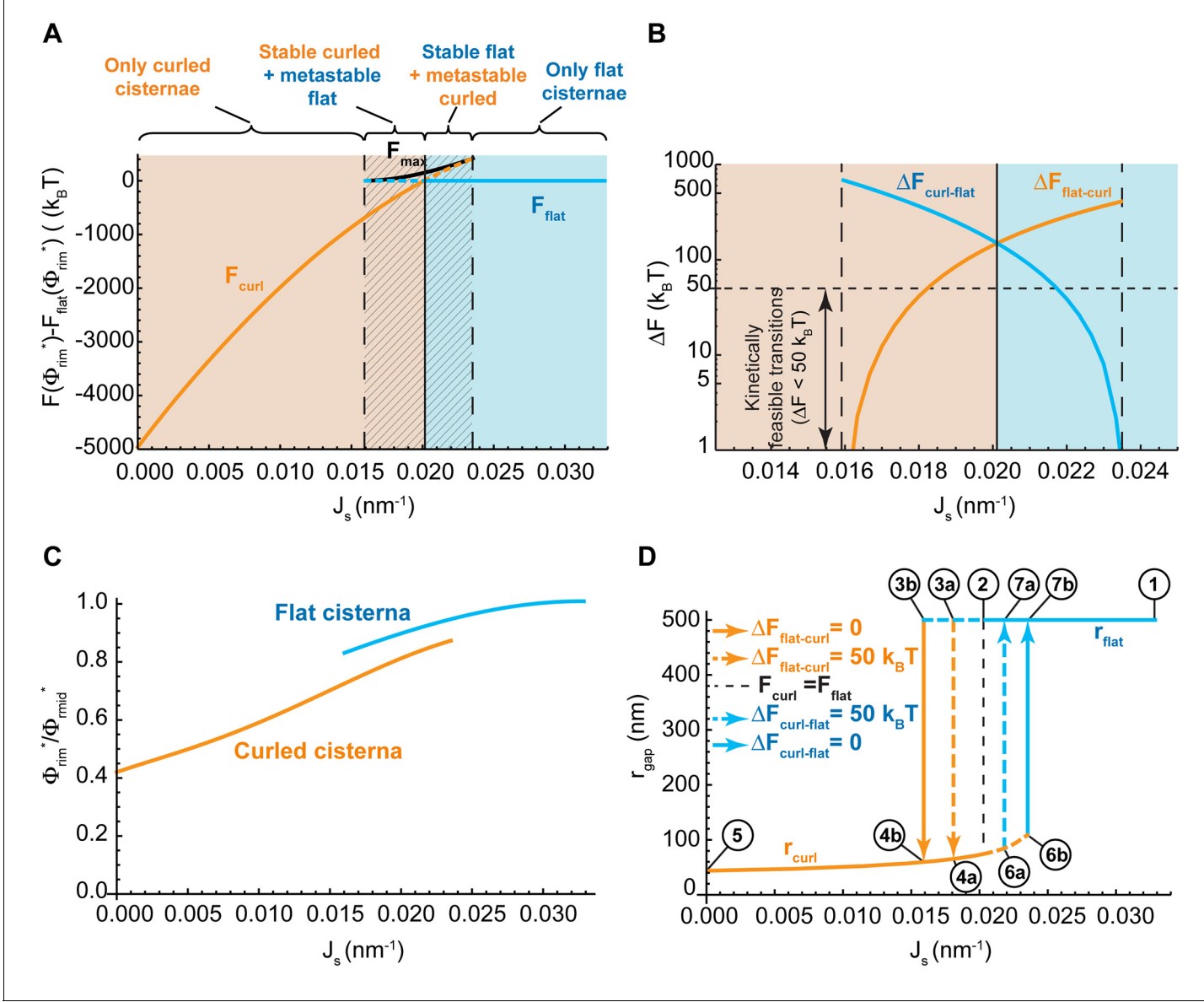

**Figure 3.** Control of the flat-to-curled cisterna transition by the membrane curvature generators. (A) The total free energy of a cisterna with respect to the flat configuration as a function of the membrane spontaneous curvature, $J_s$, for both curled (orange line) and flat (blue line) configurations. The maximum free energy of the energy barrier between flat and curled configurations, $F_{max}$, is shown as a solid black line. The different regions are color coded as in *Figure 2A*. (B) The energy barrier between curled and flat (blue line) and between flat and curled (orange line) configurations is represented in logarithmic scale as a function of the membrane spontaneous curvature, $J_s$. A horizontal dash line corresponding to a limit of kinetically feasible transitions is shown. (C) The relative enrichment in rigid nanodomains between the rim and central regions of the cisternae is represented as a function of the membrane spontaneous curvature, $J_s$, for both curled (orange line) and flat (blue line) configurations. (D) The degree of cisterna curling, $r_{gap}$, is represented as a function of the membrane spontaneous curvature, $J_s$, for both curled (orange line) and flat (blue line) configurations. Dashed lines represent metastable configurations whereas solid lines represent globally stable configurations. Different arrows representing flat-to-curled or curled-to-flat transitions are detailed in the legend. See text for details on the meaning of circled numbers. (A–D) The area fraction covered by rigid nanodomains has a fixed value of $\Phi = 0.2$.

## Hysteretic character of the cisterna shape transitions

A hallmark of bistable systems is exhibition of hysteresis in the transition between the two states (*Bhalla and Iyengar, 1999*; *Kholodenko, 2006*; *Katira et al., 2016*). This means that the system retains some kind of memory of its dynamic evolution. As a result, for the same parameter values

two different outputs can be expected depending on how the system dynamically evolved to that situation. In our case, for the same value of the membrane spontaneous curvature in the cisterna rim, $J_s$, two different cisterna configurations can form depending on how $J_s$ changed in time before reaching the final value.

To demonstrate this behavior, we considered a reversible trajectory of the system in the parameter space beginning from a situation where the flat cisterna morphology is the only stable shape (point one in the hysteresis diagram shown in *Figure 3D*) to a situation where only curled cisternae represent the locally stable shapes (point five in the hysteresis diagram shown in *Figure 3D*). For the sake of simplicity, we considered a trajectory where the nanodomain area fraction is kept constant and equal to $\Phi = 0.2$. Upon a gradual reduction of the value of $J_s$, the flat morphology ceases to be the global minimum of the free energy (point two in *Figure 3D*). However, a large energy barrier kinetically traps the system in the flat configuration, preventing it from acquiring its preferred curled morphology (*Figure 3B*). Further reduction of $J_s$ diminishes the energy barrier until the transition from a flat to a curled cisterna starts to be kinetically feasible (point 3a in *Figure 3D*) and even further to a point where the flat configuration does not correspond to a local energy minimum so that the curled state is the only equilibrium state of the system (point 3b in *Figure 3D*). The transition of the Golgi cisterna from the flat to the curled configuration is stochastically triggered at some point of the trajectory between the two values of the spontaneous curvature indicating the boundaries of the bistability region (*Figure 3D*). Once the system is out of the bistability region, further reduction of the membrane spontaneous curvature only subtly changes the shape of the curled cisternae in a continuous and smooth manner (until point five in *Figure 3D*). In the inverse process where the value of membrane spontaneous curvature, $J_s$, changes back to the initial value (point one in *Figure 3D*), the cisterna shape will remain in a curled configuration until $J_s$ reaches large enough values within the bistability region for which the curled-to-flat shape transition becomes kinetically feasible and the Golgi cisternae abruptly flattens (somewhere between points 6a and 6b in *Figure 3D*).

In summary, the results shown in *Figure 3D* indicate that, in certain conditions, upon recovery of the amount of the curvature generators present at the Golgi membranes, the shape of the Golgi cisternae can be kinetically trapped in a curled morphology, different from the initial flat shape.

## Effect of the rigid membrane nanodomains on the energy barriers of cisterna shape transitions

Next, we examined in more detail how a reduction in the amount of nanodomains at the Golgi membranes contributes to the Golgi cisterna shaping. Our model predicted that a reduction in the membrane area fraction covered by the nanodomains, $\Phi$, does not affect the flat-to-curled Golgi cisterna transition (see *Figure 2A*). Nevertheless, within the bistability region, a reduction in $\Phi$ reduces the energy barriers for both flat-to-curl and curl-to-flat transitions (*Figure 2A*, right panel). To quantify the extent of this effect, we considered two different values of the spontaneous curvature of the rim, $J_s$, within the bistability region in *Figure 2A* and computed the dependence of the transition energy barrier on $\Phi$. The results, presented in *Figure 2—figure supplement 2*, show that both the energy barrier required to flatten a curled cisterna, $\Delta F_{curl-flat}$, and the energy barrier required to curl a flat cisterna, $\Delta F_{flat-curl}$, increase with the amount of rigid nanodomains on the Golgi membranes.

## Dependence of the Golgi cisternae shape diagram on the nanodomain size and cisterna surface area

One of the parameters used to compute the shape diagram in *Figure 2A* is the size of the SM-enriched rigid lipid nanodomains, $R_d$ (see *Table 1* and Materials and methods). In order to quantify the sensitivity of our results to the value of this parameter, we computed the Golgi cisterna shape diagram for two extreme values of the nanodomain size, $R_d = 20\ nm$ and $R_d = 2\ nm$, respectively. The results, shown in *Figure 4*, indicate that the effect of the nanodomain size in controlling Golgi cisterna shape is relatively minor, and it mainly plays a role in controlling Golgi cisterna morphology by increasing the sensitivity of the shape transition to the nanodomain area fraction (compare *Figure 2A* with *Figure 4A and B*). Moreover, we also computed the shape diagram for a cisterna of four times larger surface area (two fold larger cisterna radius, $r_{flat} = 1000\ nm$). The results of the model for this situation, shown in *Figure 4—figure supplement 1*, indicate that the effect of a larger cisterna surface area on the shape diagram is in shifting the transition curves to higher values of the

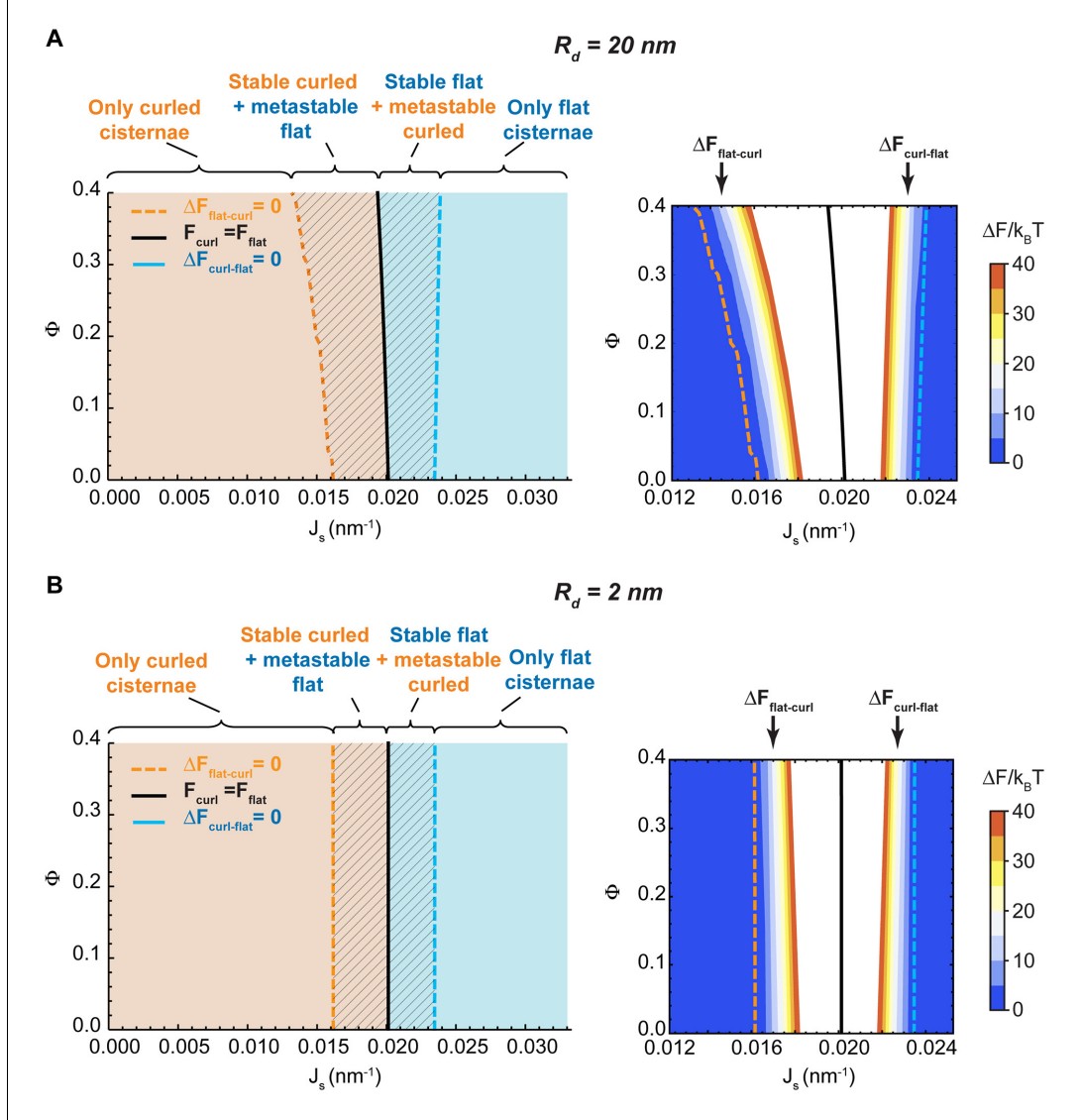

**Figure 4.** Effect of the nanodomain size on the Golgi cisterna shape diagram. (A,B) Shape diagrams for different values of the area fraction covered by nanodomains, $\Phi$, and of the membrane spontaneous curvature, $J_s$, (left panels), for two different values of the nanodomain radius, $R_d = 20$ $nm$, (A) and $R_d = 2$ $nm$, (B). Four regions can be distinguished: a region where curled cisternae are the only locally stable shapes (orange), a region with only flat cisternae (blue), and two regions where curled and flat configurations are respectively stable and metastable (orange dashed) or metastable and stable (blue dashed). The designated orange, black and blue lines indicate the boundaries between these regions. In the right subpanels, the energy barriers for the flat-to-curled or curled-to-flat transitions within the bistability regions are shown and color-coded (only energy barriers smaller than 40 $k_BT$ are shown for clarity).

The following figure supplement is available for figure 4:

**Figure supplement 1.** Effect of the cisterna surface area on the Golgi cisterna shape diagram.

spontaneous curvature of the cisterna rim. However, changing the surface area of the Golgi cisterna does not increase the sensitivity of flat-to-curled transitions to the area fraction covered by nanodomains, $\Phi$ (*Figure 4—figure supplement 1*).

## Evaluation of the effect of diacylglycerol on the cisterna shape transition

Addition of short-chain ceramide to HeLa cells leads to a local change in lipid homeostasis at the Golgi membranes (*Duran et al., 2012*). Besides the reduction in the levels of long-chain SM and the concomitant increase in short-chain SM levels, this treatment also causes an increase in the amounts of diacylglycerol (DAG) at the Golgi membranes by 30% (from 1.4 to 1.8 mol%) (*Duran et al., 2012*). DAG is a lipid characterized by an effective molecular spontaneous curvature having a large negative value, $\zeta_{DAG} \simeq -1 \ nm^{-1}$ (*Szule et al., 2002*; *Leikin et al., 1996*), and exhibiting a very fast flip-flop rate (*Ganong and Bell, 1984*; *Bai and Pagano, 1997*). The latter allows DAG molecules to homogeneously redistribute between the two membrane monolayers unless an active mechanism imposing DAG inter-monolayer asymmetry exists.

Could it be that an increase in the levels of DAG provides an alternative mechanism for the observed morphological changes of the Golgi complex? One possibility is that DAG partitions non-homogeneously within each monolayer of the Golgi cisterna membrane in such a way that the DAG distribution in the cytosolic monolayer has an opposite character to that in the luminal monolayer. Specifically, the top part of the cytosolic monolayer is enriched in DAG at the expense of the bottom part of this monolayer, whereas the bottom part of the luminal monolayer gets enriched in DAG at the expense of its top part. This DAG partitioning, which keeps the overall inter-monolayer symmetry unchanged (*Figure 5A*), might help stabilize the curvature of the sheet part of the cisterna, hence, reducing the overall bending energy of the curled state (*Figure 5A*). However, such a non-homogeneous DAG distribution along the membrane monolayers is entropically unfavorable and its extent, as well as its effect on the cisterna shape, has to be determined by minimization of the total free energy of the system accounting for both the bending elastic energy and the entropic contributions (see Materials and methods for details). In order to quantitatively evaluate these effects, we assumed that the total amount of DAG is symmetrically distributed between the luminal and cytosolic monolayers of the Golgi membrane. We numerically computed the cisterna configurations corresponding to a minimum of the total free energy (see Materials and methods, *Equation 21*) for a wide range of both the DAG mole fraction, $\phi_{DAG}$, within the system, $0 \leq \phi_{DAG} \leq 0.05$, and of the spontaneous curvature of the cisterna rim membrane, $0 \leq J_s \leq 0.033 \ nm^{-1}$. The results, presented in *Figure 5B*, show that there is a very weak dependency of the preferred cisterna shape on DAG levels. Moreover, our theoretical model predicts that, for the experimentally observed increase of DAG levels from 1.4% to 1.8% (*Duran et al., 2012*), would not lead to a morphological transition of a flat cisterna to the curled form (*Figure 5B*). Finally, we computed the relative distribution of DAG amongst the different monolayers of the Golgi cisterna. These results show that the DAG distribution is almost homogeneous for all the cisterna monolayers (*Figure 5C*), due to the relatively large entropy cost of inhomogeneous partitioning of such small molecules, in agreement with previous studies (*Derganc, 2007*; *Sorre et al., 2009*). Taken together, these results suggest that the increase in the levels of DAG is not the driving force for the observed curling of the Golgi cisternae.

## Experimental results

### Short-chain ceramide treatment causes the release of clathrin coats from the Golgi membranes prior to cisterna curling

One of the main predictions of our model is that the flat-to-curled cisterna transition results from a reduction in the amounts of curvature generators present at the Golgi membrane (*Figure 2A*). We decided to experimentally test this prediction by investigating whether short-chain ceramide treatment, which specifically disrupts SM homeostasis at the Golgi membranes and is known to drive formation of curled Golgi cisternae (*van Galen et al., 2014*), induces the release of peripheral membrane proteins implicated in generation of membrane curvature in the course of transport carrier formation at the Golgi membranes. It has been recently found that the metabolic generation of SM at the Golgi membranes results, via a DAG-triggered signaling pathway, in the local reduction of PI(4)P levels causing the release of PI(4)P-binding proteins, such as the clathrin-adaptor protein γ-adaptin, without affecting the localization of COPI components (*Capasso et al., 2017*). Hence, we investigated whether short-chain ceramide treatment induces the release of endogenous clathrin heavy chain (clathrin-HC), one of the components of clathrin-coated vesicles, the subunits of which polymerize into a cage-like triskelion structure involved in bending the underlying membrane (*Kirchhausen, 2000*). To this end, HeLa cells were treated with short-chain ceramide (D-cer-C6, 20 μM) for

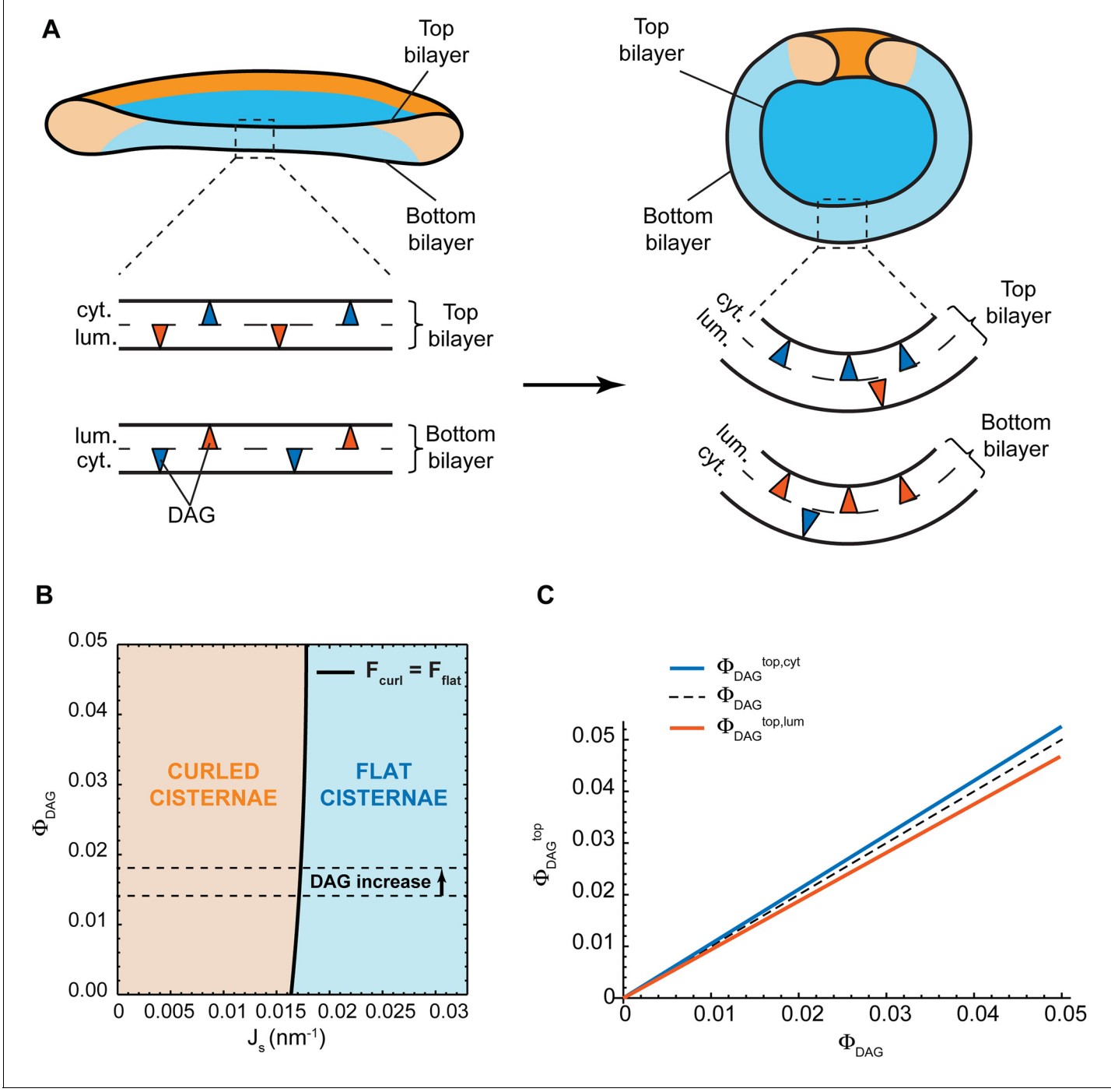

**Figure 5.** Effect of lateral DAG partitioning on the cisterna shape transition. (**A**) Schematic representation of the proposed mechanism of non-homogeneous DAG partitioning along the two monolayers of a Golgi cisterna for both flat (left) and curled (right) configurations. Colored triangles represent DAG molecules in the cytosolic (blue) and luminal (red) leaflets of both the top and bottom bilayers of the Golgi membrane. (**B**) Shape diagram showing the globally stable configuration of the system (curled cisterna in orange, flat cisterna in blue) for different values of the total molar fraction of DAG in the membrane, $\Phi_{DAG}$, and of the membrane spontaneous curvature, $J_s$. The experimentally observed increase in DAG levels after short-chain ceramide treatment is indicated by the arrow between the two dashed black lines. (**C**) Optimal molar fraction of DAG in the cytoplasmic ($\Phi_{DAG}^{top, cyt}$, blue line) and luminal ($\Phi_{DAG}^{top, lum}$, red line) monolayers of the cisterna top bilayer plotted as a function of the total molar fraction of DAG, $\Phi_{DAG}$. The dashed line corresponds to a homogeneous DAG distribution.

different times, after which the cells were fixed and the localization of endogenous clathrin-HC and the TGN protein p230 was monitored by immunofluorescence microscopy. We observed that, already after 30 min of treatment with short-chain ceramide, the pool of clathrin-HC initially present at the Golgi membranes was mostly released (*Figure 6A,B*). These results indicate that, indeed, the formation of curled cisternae upon the treatment with short-chain ceramide proceeds in parallel with a decrease in the amounts of curvature inducers at the Golgi membranes.

We next assessed the relative timing of the release of the clathrin coats from the Golgi membranes with respect to the flat-to-curled Golgi cisternae transition induced by short-chain ceramide treatment. For this purpose, we analyzed the ultrastructural morphology of the Golgi complex by immuno-electron microscopy after different times of short-chain ceramide treatment. Our results show that Golgi cisternae curling starts already after 30 min of D-cer-C6 treatment, and that after 4 hr of treatment, virtually no flat stacks are observed (*Figure 6C,D*). In agreement with our previous analysis (*van Galen et al., 2014*), cisternae curling occurs towards the *trans*-Golgi cisternae/TGN, as monitored by the localization of specific late Golgi markers (*Figure 6C*). We extracted from these images the radius of flat cisternae, $r_{flat} = 510 \pm 21$ *nm* (average $\pm$ SEM; N = 22) and the radius of curling in the 4 hr D-cer-C6 treated Golgi cisternae, $R = 270 \pm 26$ *nm* (average $\pm$ SEM; N = 14), which are in agreement with the condition of area conservation (*Equation A2*). Taken together, these results indicate that short-chain ceramide treatment leads to the release of clathrin coats thus reducing the spontaneous curvature of the Golgi membranes, which, as our model predicts, leads to the curling of the Golgi cisternae.

## Experimental evidence for hysteresis of Golgi cisternae morphology during recovery from short-chain ceramide treatment

Our model predicts the existence of a bistability region in the cisterna shape diagram, a relatively large range of parameters where both flat and curled Golgi cisternae correspond to locally stable shapes (*Figure 2A*). As we showed, the transitions between the flat and curled configurations within the bistability region are expected to have a hysteretic character (*Figure 3D*). To experimentally test whether Golgi cisterna shape transition induced by short-chain ceramide treatment exhibits hysteresis, we performed short-chain ceramide washout experiments to monitor the timing of recovery of both the Golgi morphology and the amounts of clathrin present at the Golgi membranes as the cells return to steady conditions.

We first investigated how the dynamics of recovery of the Golgi cisternae shape during short-chain ceramide washout correlates with the recruitment of clathrin coats to the Golgi membranes. To this aim, we pre-treated HeLa cells with D-cer-C6 for 30 min, after which the cells were extensively washed and incubated with normal growth medium for different times. Then the cells were fixed and the intracellular localization of clathrin-HC was monitored by immunofluorescence microscopy. The results of this experiment show that a 2 hr short-chain ceramide washout is sufficient to recover similar levels of clathrin-HC at the Golgi membranes as to those found in untreated cells (*Figure 6A,B*). Recovery of normal clathrin-HC levels at the Golgi membranes after a 4 hr D-cer-C6 treatment is slower, and occurs in about 6 hr (*Figure 6—figure supplement 1*).

Next, we performed analogous washout experiments to monitor the timing of Golgi cisternae shape recovery. We added D-cer-C6 to HeLa cells for 30 min, after which the cells were extensively washed and incubated in complete medium without D-cer-C6 for different times. The cells were then fixed and the ultrastructure of the Golgi cisternae was visualized by immuno-electron microscopy (*Figure 6C*). Our results show that the curled-to-flat Golgi cisterna transition during short-chain ceramide washout occurs at a much slower kinetics as the recovery of the clathrin coats to the Golgi membranes (*Figure 6B,D*). Indeed, even 16 hr after the short-chain ceramide washout –a condition where the Golgi membranes already recovered their stationary pools of clathrin coats– both flat and curled Golgi cisternae can still be observed (*Figure 6C,D*).

To confirm these observations, we used an alternative approach to quantitate the dynamics of the curled-to-flat Golgi shape transition promoted during short-chain ceramide washout. We took advantage of the fact that the curling of the Golgi cisternae induced by short-chain ceramide is accompanied by a lateral segregation of different Golgi-resident proteins, such as TGN46 and p230, and that the level of this segregation can be quantitatively assessed by immunofluorescence microscopy (*van Galen et al., 2014*). Although the observed protein segregation correlates with changes

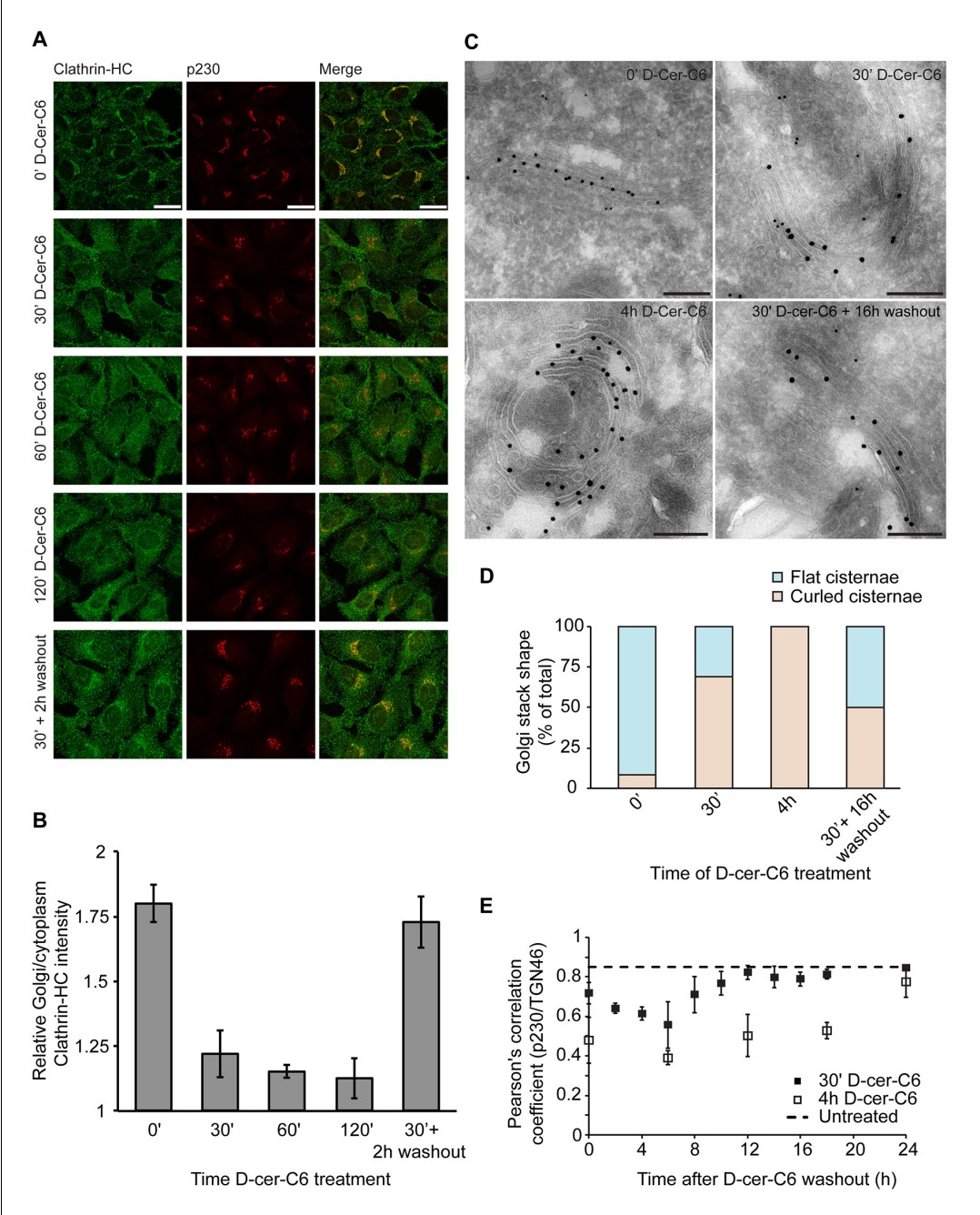

**Figure 6.** Experimental results. (**A**) HeLa cells were treated with 20 μM D-cer-C6 for the indicated times, after which the cells were fixed and the localization of clathrin-HC and the TGN marker protein p230 was monitored by immunofluorescence microscopy. Scale bar is 25 μm. (**B**) Quantitation of the results in (**A**) showing the intensity of clathrin-HC in the Golgi area relative to the intensity in the rest of the cytoplasm, for at least 15 cells from three different experiments. Bars represent average values and error bars are the S.E.M. (**C**) HeLa cells stably expressing the Golgi-resident protein Mannosidase-II-GFP were treated with 20 μM D-cer-C6 for the indicated times, fixed, and the Golgi complex ultrastructure visualized by immunoelectron microscopy. Gold particles of 10 nm and 15 nm label p230 and GFP, respectively. Scale bar is 200 nm. (**D**) Quantitation of the percentage of flat (blue) and curled (orange) cisternae in the Golgi stacks observed in the experiment presented in (**C**). (**E**) HeLa cells were treated with 20 μM D-cer-C6 for either 30' (solid black squares) or 4 hr (empty squares), after which the cells were extensively washed and incubated for different times in complete medium without D-cer-C6. Then, the cells were fixed and the levels of co-localization of two *trans*-Golgi membrane proteins p230 and TGN46 were quantitated from the immunofluorescence images by means of the Pearson's correlation coefficient, which is plotted here as a function of the washout time. The dashed horizontal line represents the Pearson's correlation coefficient for TGN46 and p230 in untreated HeLa cells.

The following figure supplement is available for figure 6:

*Figure 6 continued on next page*

*Figure 6 continued*

**Figure supplement 1.** Relative localization of Clathrin-HC at the Golgi area in D-cer-C6 treated cells.

in Golgi membrane morphology, its driving mechanisms still remain unknown (*van Galen et al., 2014*). To monitor the dynamics of protein segregation during Golgi shape recovery after short-chain ceramide treatment, HeLa cells were treated with D-cer-C6 for 30 min or 4 hr, after which the cells were extensively washed, and incubated in normal medium without short-chain ceramide for different times before being fixed. Then, the intracellular localization of the two *trans*-Golgi membrane proteins p230 and TGN46 was monitored by immunofluorescence microscopy and the relative colocalization of the two proteins was quantitated by means of the Pearson's correlation coefficient. Our results confirm that cells pre-treated with short-chain ceramide for 30 min required about 12 hr to recover the initial levels of p230 and TGN46 colocalization, whereas a longer 4 hr pre-treatment with D-cer-C6 required about 24 hr for a complete recovery (*Figure 6E*).

Taken together, these results indicate that, after short-chain ceramide washout, the recovery of the levels of clathrin-HC at the Golgi membranes (which we suggest parallels the recovery of the initial values of the membrane spontaneous curvature) occurs much faster than the recovery of the flat cisternae morphology and of protein colocalization. This is indicative of a hysteretic behavior of the transition from flat-to-curled cisternae and reverse, as our model predicts.

## Discussion

The architecture of the Golgi complex in higher eukaryotes has been the subject of extensive research using numerous experimental approaches, including electron and immunofluorescence microscopy techniques (see [*Klumperman, 2011*] for a review). Generally, the Golgi complex in mammalian cells consists of a set of 4–8 flat-like cisternae stacked to each other (*Emr et al., 2009*). Each of these cisternae has a relatively flat central part and a highly curved rim. What is the connection between the shape of Golgi cisternae and the functions of this organelle? It has been suggested that the large surface-to-volume ratio of the Golgi cisternae helps accommodating the continuous influx and efflux of transport carriers to and from these membranes (*Griffiths et al., 1989*; *Glick and Nakano, 2009*). Moreover, two of the principal functions of the Golgi complex –protein glycosylation and transport– need to be spatially and timely organized to ensure their efficiency. Thus, it has been proposed that processing events localize preferentially at the central flat part of the Golgi cisternae, whereas transport carrier formation occurs at the rims of the Golgi membranes (*Rothman, 2010*; *Popoff et al., 2011*; *van Galen et al., 2014*; *Engel et al., 2015*).

We previously showed the importance of SM homeostasis in protein organization and function at the Golgi membranes (*Duran et al., 2012*; *van Galen et al., 2014*). In particular, we revealed that selective disruption of SM organization at the Golgi membranes leads to (i) an overall reduction in the lateral order of the Golgi membranes (*Duran et al., 2012*), (ii) a strong inhibition of transport carrier formation at the Golgi complex (*Duran et al., 2012*), and (iii) a defect in the formation of functional enzymatic domains caused by the physical segregation between Golgi resident enzymes and their substrates (*van Galen et al., 2014*). Intriguingly, these effects parallel an abrupt change in the morphology of the Golgi complex, which turns from a stack of flat cisternae into an onion-like stack of highly curled cisternae (*van Galen et al., 2014*). In the present study, we aimed at resolving the mechanism by which SM metabolism controls the morphology of the Golgi cisternae. Our approach consisted in the elaboration of a theoretical biophysical model of Golgi cisternae morphology that describes the membrane free energy including the contributions arising from the elastic energy of membrane bending and the entropic contribution of lateral partitioning of SM-rich nano-domains. Our model explains the existence of two distinct families of Golgi cisternae shapes, flat and highly curled cisternae. Moreover, our model predicts the existence of a flat-to-curled shape transition triggered by a reduction in the amounts of membrane curvature generators at the Golgi membranes. We experimentally tested this theoretical prediction and observed that clathrin, a protein involved in the assembly of the large membrane curvatures required for the formation of clathrin-coated vesicles at the Golgi membranes, was reduced at the Golgi membranes as a result of SM homeostasis alteration.

Another key prediction of our model is that the flat-to-curled Golgi cisterna transition is practically insensitive to changes in the amounts of SM-rich rigid nanodomains (*Figure 2A*). These results indicate that such rigid membrane domains are not sufficient to stabilize the flat cisterna configuration in the absence of membrane curvature generating proteins stabilizing the highly curved rim (see orange region in the shape diagram, *Figure 2A*). However, our model predicts a non-homogeneous partitioning of such lipid nanodomains along the Golgi membrane (*Figure 3C*). Specifically, rigid nanodomains tend to be concentrated at the central, flatter part of the cisterna rather than at the rim. The extent of the nanodomain redistribution from the rim to the central part of the cisterna is gradually magnified upon the decrease in the amounts of curvature generating proteins at the rim for both flat and curled cisterna configurations (*Figure 3C*). We hypothesize that such redistribution could be a causal link to the previously reported defects in protein glycosylation in cells where SM homeostasis had been altered (*van Galen et al., 2014*). According to this conjecture, lateral segregation of Golgi-resident enzymes from their substrates would follow from the lateral repartitioning of lipid nanodomains along the surface of the Golgi cisterna. We propose that during short-chain ceramide washout, three processes with different dynamics occur simultaneously (see *Figure 7*): (i) recovery of normal levels of membrane curvature generators, which, for a 30' treatment, takes about 2 hr (*Figure 6B*); (ii) recovery of the amounts of nanodomains, which we propose is in direct correlation to the recovery of protein colocalization at the Golgi membranes, takes about 12 hr (*Figure 6E*); and (iii) recovery of the flat Golgi morphology, which takes about 16 hr (*Figure 6D*). Testing this hypothesis requires further experimental work, the results of which will advance our understanding of the role that lipid homeostasis and membrane lateral organization play in regulating the functions of the Golgi complex.

## DAG acts as a signaling effector rather than as a molecular shaper to control Golgi membrane morphology

DAG is a key regulator involved in transport carrier biogenesis at the Golgi membranes (*Baron and Malhotra, 2002*; *Bard and Malhotra, 2006*; *Fernández-Ulibarri et al., 2007*; *Malhotra and Campelo, 2011*). On the one hand, DAG is a conical lipid, which has the ability on its own to generate negative (positive) membrane curvature if asymmetrically enriched in the cytosolic (luminal) leaflet of a membrane (*Carrasco and Mérida, 2007*; *Leikin et al., 1996*; *Szule et al., 2002*). On the other hand, protein kinase D (PKD), a protein that controls the fission of TGN-to-cell surface transport carriers (*Malhotra and Campelo, 2011*; *Campelo and Malhotra, 2012*), is directly recruited to the TGN by binding to DAG (*Maeda et al., 2001*; *Liljedahl et al., 2001*; *Baron and Malhotra, 2002*). It has been reported that low levels of DAG leads to defects in protein export from the TGN (*Baron and Malhotra, 2002*) and to abnormal Golgi morphology (*Litvak et al., 2005*). In contrast, local increase in DAG levels leads to the activation of PKD (*Malhotra and Campelo, 2011*). Active PKD phosphorylates a number of substrates at the TGN membranes, including the lipid kinase PI4KIII$\beta$ (*Hausser et al., 2005*), the lipid transport proteins CERT (*Fugmann et al., 2007*), and OSBP (*Nhek et al., 2010*), thus regulating the local lipid homeostasis. The theoretical results presented here (*Figure 5*) indicate that changes in the DAG levels do not directly promote a flat-to-curled Golgi cisternae transition by redistributing along the membrane and changing its curvature. Importantly, it has been recently shown that a local burst in DAG levels at the Golgi membranes caused by increased sphingolipid metabolic flow leads to a peak of activation of PKD, which in turn, through a downstream signaling cascade, results in the consumption of PI(4)P and the consequent release of PI (4)P-binding proteins (*Capasso et al., 2017*). Phosphoinositides are mostly localized away from SM-rich rigid nanodomains (*Arumugam and Bassereau, 2015*), thus suggesting that this signaling event occurring at the cytosolic side of the membrane is spatially uncoupled from SM-rich rigid nanodomains in the lumenal leaflet. Based on these results and on our results showing that clathrin coats are released from the Golgi membranes prior to short-chain ceramide-mediated flat-to-curled Golgi cisterna transition, we propose that SM metabolism, through the by-product DAG, indirectly controls Golgi morphology by means of a downstream PKD-dependent signaling cascade rather than by playing a direct mechanical part in membrane bending.

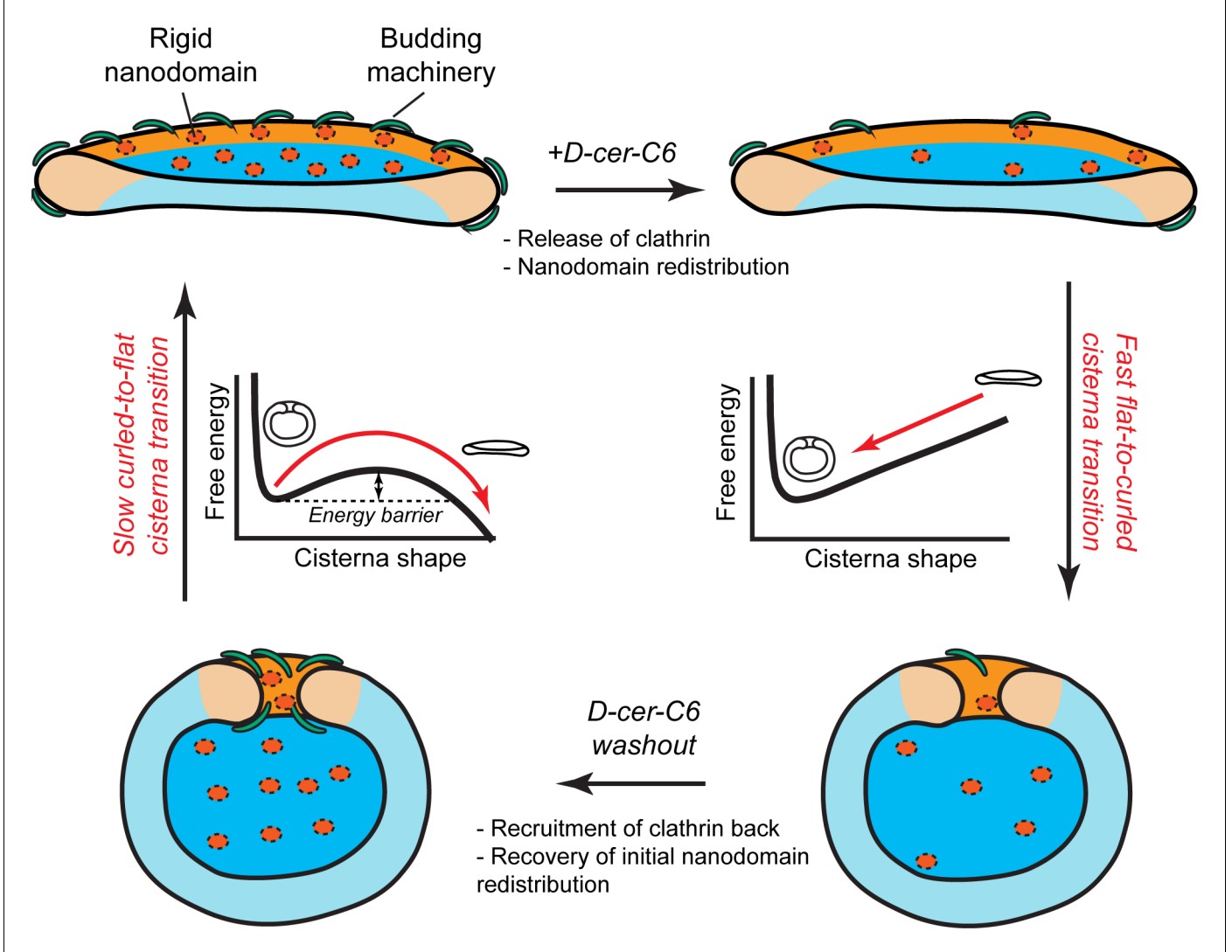

**Figure 7.** Model of how SM metabolism controls the shape of a Golgi cisterna. In stationary conditions (top left cisterna), a Golgi cisterna appears as a flat, disc-like structure, with relatively large amounts of budding machinery (green proteins), such as the components of the clathrin-coated vesicles. The Golgi membranes contain about 10% molar fraction of long chain SM, which could be organized in small rigid nanodomains (red patches). Treatment of cells with D-cer-C6 leads to a reduction in the levels of rigid domain-forming SM, a release of clathrin-HC from the Golgi membranes and, according to our physical model, a lateral redistribution of the remaining rigid nanodomains away from the rim (top right cisterna). Under these conditions, the cisterna free energy profile has a single minimum corresponding to a highly curled cisterna configuration and hence, a rapid flat-to-curled cisterna transition is promoted (bottom right cisterna). Washout of D-cer-C6 leads to the recovery of stationary levels of clathrin to the Golgi membranes and, we hypothesize, of the initial levels of rigid SM-rich nanodomains (bottom left cisterna). Under these conditions, the cisterna free energy profile has two local minima corresponding to highly curled and flat cisterna configurations, separated by an energy barrier. The system can thus be kinetically trapped in the curled, metastable configuration and therefore a slow transition back to the flat configuration (top left cisterna) is expected.

## Membrane curvature generators dynamically stabilize the flat shape of Golgi cisternae

One of the main results of our model is that the release of membrane curvature-generating proteins leads to the destabilization of the flat Golgi cisterna configuration, triggering a morphological transition towards a curled configuration. Being the central hub of the secretory pathway, the Golgi complex recruits a number of different curvature-generating proteins to efficiently sustain transport carrier formation (*Cruz-Garcia et al., 2013*; *Bonazzi et al., 2005*; *Campelo and Malhotra, 2012*).

Our results suggest that the role of such proteins is twofold. First, they induce membrane curvature to accommodate the secretory cargoes into nascent budding carriers prior to their fission from the Golgi membranes. Second, this dynamic series of budding and scission events serves to stabilize the highly bent rims of the Golgi cisternae. As such, we propose that the shape and function of Golgi membranes are maintained by membrane curvature generators via a positive feedback loop where the highly bent Golgi rims provide optimal nucleation sites for the budding of transport carriers. At the same time, the machinery involved in this process maintains and stabilizes a functionally optimal Golgi cisternae morphology. Does the presence of clathrin coats on the Golgi membranes represent the main driving force for flat cisternae stabilization? Or are there other curvature-inducing proteins involved? Altered SM metabolic flow at the Golgi membranes did not affect the localization of COPI components (*Capasso et al., 2017*) to those membranes. However, other curvature generators might be released in addition to clathrin due to the defects in SM metabolism. It has been recently reported that knockdown of the two Golgi-localized PI(4) kinases in Atg5 knockout cells induces curling of the Golgi cisternae (*Yamaguchi et al., 2016*). Moreover, it has been shown that components of the COPI machinery are released from the Golgi membranes in HeLa cells incubated at 15°C, a situation that parallels curling of the Golgi cisternae towards the *cis* side of the stack (*Martínez-Alonso et al., 2005*). Altogether, we propose that maintenance of the flat Golgi cisternae morphology requires a combined effort of different classes of curvature generating proteins and that the release of a subset of these proteins can lead to the destabilization of the flat configuration. We suggest that the breaking of the stack symmetry upon cisternae curling is driven by the initial release of rim stabilizers from a few cisternae (*trans*-cisternae in our experiments, *cis*-cisternae in [*Martínez-Alonso et al., 2005*]), which is then followed by the other cisternae. From a theoretical perspective, the effect of including multiple cisternae with different levels of budding effectors is analyzed and discussed in the Appendix. In brief, our results indicate that Golgi curling must parallel some level of release of curvature generators from all Golgi cisternae. Further experiments are needed to test this.

Our model predicts the existence of shape bistability within a certain range of values of the membrane spontaneous curvature and of the membrane area fraction covered by rigid nanodomains (*Figure 2A*). Within this region of the parameter space, both flat and curled cisterna configurations correspond to locally stable shapes (*Figure 2B*). This means that the system can be kinetically trapped in a metastable configuration, which corresponds to a local but not global minimum of the cisterna free energy before it relaxes to the globally stable configuration. Such transition from a metastable to a stable configuration needs to overcome an energy barrier. If the value of the energy barrier is relatively small, transition to the stable configuration can be overcome by thermal fluctuations of the Golgi cisterna shape. Such thermally-triggered transitions follow Arrhenius kinetics, according to which an average transition time, $\tau$, can be estimated as $\tau = t_0 e^{\Delta F/k_B T}$, where $t_0$ is a characteristic time scale of Golgi cisterna fluctuations and $\Delta F$ is the height of the energy barrier (*Hänggi et al., 1990*; *Morlot et al., 2012*). As mentioned above, we can estimate $t_0 \approx 1\ ms$ from hydrodynamic arguments as $t_0 = \eta R^3/\kappa$ (*Allain et al., 2004*), where $\eta$ is the cytosol viscosity, $R$ is a typical length scale of a Golgi cisterna, and $\kappa$ is the bending rigidity of the membrane. We can qualitatively compare this to our experimental results on the curled-to-flat Golgi cisterna transition during short-chain ceramide washout (*Figure 6*). Those results indicated that 14 hr after the recovery of normal clathrin levels at the Golgi membranes, about 50% of Golgi stacks were composed of flat cisternae (*Figure 6D*). According to the aforementioned kinetics, such shape transition time corresponds to an energy barrier for the curled-to-flat transition of $\Delta F \approx 18\ k_B T$. If we compare this estimation for the curled-to-flat transition to the numerical results of our model (*Figure 2A*), we find that the spontaneous curvature at the Golgi cisterna rims in the fully recovered state should be of the order of $J_s \approx 0.0225\ nm^{-1}$. Following the relationship between the membrane spontaneous curvature and the area fraction covered by the curvature generators (*Campelo et al., 2008*), the predicted cisterna spontaneous curvature corresponds to a membrane area fraction covered by curvature generators of $5 - 10\%$, which is a physiologically reasonable estimation. Moreover, we showed that the time of recovery of Golgi protein localization after short-chain ceramide treatment depends on the duration of the treatment (*Figure 6E*). These different recovery times, we suggest, can be explained by the fact that the longer the treatment with short-chain ceramide, the lower the levels of long-chain SM at the Golgi membranes. Hence, a longer time would be required for the Golgi membranes to

recover their normal levels of SM-rich nanodomains and, according to our hypothesis, of Golgi protein recovery.

Likewise, our model predicts that the Golgi cisterna shape transition is, in thermodynamic terms, a first-order transition because the transition is discontinuous in the shape parameter (in our case, the distance between the center of the cisterna rim and the axis of symmetry, $r_{gap}$) (*Figure 3D*). This indicates that once the transition from a flat cisterna ($r_{gap} = 500 \ nm$) to a highly curled cisterna ($40 \ nm < r_{gap} < 100 \ nm$) is triggered, the transition is abrupt because no cisternae of intermediate curling correspond to a locally stable configuration (*Figure 3D*). Although it is hard to extract quantitative information of the curled cisterna gap opening size, $r_{gap}$, from the ultrathin sections (*Figure 6C*) to compare with the theoretical predictions, our ultrastructural analysis of the Golgi morphology qualitatively showed that Golgi curling indeed occurs in an abrupt manner (*Figure 6C*).

In summary, we have presented a theoretical biophysical model of Golgi cisterna morphology, which describes the existence of stable flat and curled Golgi cisternae for different values of the membrane spontaneous curvature. We experimentally validated some of the model's predictions. In particular, our model helps explaining the mechanisms by which a reversible flat-to-curled Golgi cisternae transition is induced upon disruption of SM homeostasis by short-chain ceramide treatment. Flat Golgi cisternae in untreated HeLa cells have stationary levels of different curvature-inducing proteins, such as components of the clathrin-coated vesicle machinery (*Figure 7*, top left cartoon). Moreover, a certain amount of small, dynamic, SM-enriched rigid nanodomains might be present in the membrane, and slightly enriched in the central flat part of the cisterna (*Figure 7*, top left cartoon). Treatment of cells with D-cer-C6 has a twofold effect on the Golgi membrane properties: it causes the release of clathrin from the membranes (*Figure 6A,B*) and decreases the lateral order of the Golgi membranes (*Duran et al., 2012*) (*Figure 7*, top right cartoon). The results of our model show that the decrease in membrane spontaneous curvature (through the release of curvature generating proteins such as clathrin) but not a reduction in the number of rigid nanodomains alters the cisterna free energy profile to a situation where the flat cisterna configuration is unstable and hence a fast flat-to-curled cisterna transition occurs (*Figure 7*, right). Short-chain ceramide washout leads to the recruitment of clathrin back to the Golgi membranes (*Figure 6A,B*) and, we suggest, also leads to the recovery of the initial levels of SM-enriched rigid nanodomains (*Figure 7*, bottom left). Under these conditions the system free energy profile presents shape bistability, so the Golgi cisternae are kinetically trapped in the curled configuration. Hence, the curled-to-flat cisterna transition is slow because it requires the energy barrier to be overcome by thermal fluctuations (*Figure 7*, left).

Overall, the model presented in here together with some of its experimental validation underscore the crucial role of SM metabolism in regulating the structural morphology and function of the Golgi cisternae. We foresee that future experimental work along these lines will strengthen our predictions and will help to understand better the different factors governing the shape and function of the Golgi complex.

## Materials and methods

### Reagents and antibodies

*N*-hexanoyl-D-erythro-sphingosine (D-cer-C6) was obtained from Matreya and dissolved in pure ethanol (Merck) to a 10 mM stock solution. Sheep anti–human TGN46 was obtained from AbD Serotec (Bio-Rad / AbD Serotec Cat# AHP500, RRID:AB_324049). Mouse anti-p230 was obtained from BD (BD Biosciences Cat# 611280, RRID:AB_398808). Goat anti-Clathrin-HC antibody was from Santa Cruz (Santa Cruz Biotechnology Cat# sc-6579, RRID:AB_2083170). Alexa Fluor–labeled secondary antibodies were obtained from Invitrogen.

### Cell culture

HeLa cells, obtained from ATCC, were cultured in DMEM (Lonza) containing 10% FCS. HeLa cells stably expressing the plasmid encoding the first 100 amino acids of rat mannosidase-II in the pEGFP-N1 vector (HeLa-MannII-GFP cells) were described previously (*Sütterlin et al., 2005*; *van Galen et al., 2014*). All cell lines were periodically checked for mycoplasma contamination.

## Immunofluorescence microscopy

For clathrin-HC immunostaining, samples were fixed and permeabilized in methanol for 6 min at −20°C. For p230 immunostaining, samples were fixed with 4% formaldehyde in PBS for 20 min and permeabilized with 0.2% Triton X-100 in PBS for 30 min. Fixed cells were then blocked in 2% BSA in PBS for 30 min before antibody staining. Cells were then sequentially incubated for 1 hr at room temperature first with primary and then with secondary antibodies diluted in blocking buffer. Samples were analyzed with a confocal system (TCS SP5 II CW STED; Leica) in confocal mode using a 100x, 1.4 NA objective and HyD detectors (Leica). Alexa Fluor 488–, 568-, 594-conjugated secondary antibodies were used. Images were acquired using the Leica software and converted to TIFF files using ImageJ (version 1.43; National Institutes of Health). Two-channel colocalization analysis was performed using ImageJ, and the Pearson's correlation coefficient was calculated using the Manders' coefficients plugin developed at the Wright Cell Imaging Facility (Toronto, Ontario, Canada).

## Immunoelectron microscopy

The samples were fixed and prepared using standard methods, essentially as described previously (*Rizzo et al., 2013*). In brief, the cells were fixed with 2% paraformaldehyde and 0.2% gluteraldehyde in PBS, for 2 hr at room temperature. The cells were then washed with PBS/0.02 M glycine, scraped in 12% gelatin in PBS, and then embedded in the same solution. The cells embedded in gelatin were cut in 1 mm blocks and infiltrated with 2.3 M sucrose at 4°C, mounted on aluminum pins, and frozen in liquid nitrogen. The samples were then sectioned and the ultrathin cryosections were picked up in a mixture of 50% sucrose and 50% methylcellulose and incubated with antibodies to antigen of interest (anti-GFP and anti-p230) followed by protein A gold. The samples were observed in the FEI Tecnai-12 electron microscope.

## Physical model of Golgi cisternae morphology

In this section we formulate in mathematical terms the physical model we used to describe how Golgi cisterna morphology is controlled by variations in SM homeostasis and metabolism. In the two following subsections we describe, respectively, (i) the system geometry, that is, the possible geometrical configurations of the Golgi cisterna; and (ii) the system free energy.

### Geometrical description of a Golgi cisterna

The Golgi complex in mammalian cells and in most eukaryotes consists of multiple stacks of flattened disc-like cisternae (*Klumperman, 2011*). Our previously (*van Galen et al., 2014*) and presently reported ultrastructural data shows that normal Golgi morphology is altered in cells where SM metabolism had been disrupted. Under those conditions, the original flat-like cisternae curl into a concentric stacked onion-like structure. Based on those data, we describe the morphology of a single flat Golgi cisterna as explained in the main text (*Figure 1C*). In the Appendix we describe the effect of having multiple stacked cisternae. The overall Golgi curling is geometrically characterized by the radius of curvature of the Golgi cisterna, $R$, which tends to infinity for a completely flat Golgi cisterna (*Figure 1C*). Alternatively, the degree of Golgi curling can also be described by the distance between the center of the cisterna rim and the axis of symmetry, $r_{gap}$ (*Figure 1C*). Hence, $r_{gap} = r_{flat}$ for a completely flat cisterna, and $r_{gap} \to r_{rim}$ for highly curled cisternae. Based on the ultrastructural data, we assume that the total surface area of a Golgi cisterna membrane, $A$, the luminal thickness of the cisterna, $2h$, and the radius of the rim, $r_{rim}$, do not change as a result of cisternae deformation (*van Galen et al., 2014*). In an unconstrained system, the natural tendency of the system to minimize the bending energy is by adjusting the radius of the rim to match the spontaneous curvature. In our model, the radius of the rim cross-section is set by the distance between a protein scaffold forming the flat part of a cisterna and the cisterna edge. For example, if such scaffold would extend beyond the cisterna, e.g. if a cisterna would be formed by flattening of a big liposome between two infinite flat rigid plates, the edge cross-sectional radius would be simply equal to the cisterna half-thickness, independently of the spontaneous curvature of the rim. The fact that the rim is somewhat swollen as compared to the cisterna thickness reflects the distance to which the scaffold edge approached the cisterna edge. In this model, the membrane spontaneous curvature could influence the detailed shape of the edge cross-section profile resulting in its deviation from the circular shape. This would be the case for the shapes where $r_{rim}$ is comparable to $r_{gap}$. Taking into account this effect would

result in corrections of the rim energy but on a semi-quantitative level of description we neglect these corrections. The molecular identity of such protein scaffold could be the spacer proteins maintaining the luminal thickness and/or the stacking factors keeping the subsequent cisternae stacked. Interestingly, it has been experimentally shown that knockdown of the stacking/tethering factors GRASP55/65 or Golgin45/GM130 in HeLa cells leads to the cisternae unstacking, and swelling of the lumen and rims of the Golgi cisternae (*Lee et al., 2014*). Altogether this means that the total area, $A = A(r_{gap})$, which is the sum of the area of the central part of the cisterna, $A_{mid}(r_{gap})$, and the area of the rim, $A_{rim}(r_{gap})$, is the same regardless of the cisternae morphology, that is, for all values $r_{rim} < r_{gap} \leq r_{flat}$. On the one hand, the total surface area of the rim region, $A_{rim}(r_{gap})$, can be obtained by using the area element,

$$dA_{rim} = r_{rim}(r_{gap} + r_{rim}\cos\varphi)d\varphi d\theta, \tag{1}$$

where $\{\varphi, \theta\}$ are toroidal coordinates (see *Figure 1—figure supplement 1*). Then, the surface area of the rim is written as

$$A_{rim} = \iint dA_{rim} = r_{rim}\int_0^{2\pi} d\theta \int_{\varphi_0 - \frac{\pi}{2} + \alpha}^{\varphi_0 + \frac{3\pi}{2} - \alpha} (r_{gap} + r_{rim}\cos\varphi)d\varphi = 4\pi r_{rim}$$

$$\left[r_{gap}(\pi - \alpha) - r_{rim}\sin\alpha\,\sin\varphi_0\right], \tag{2}$$

where $\varphi_0 = \arcsin(1 - 2r_{gap}^2/r_{flat}^2)$ and $\alpha = \arcsin(h/r_{rim})$ (see *Figure 1—figure supplement 1*). On the other hand, we can express the area of the central part of the cisterna as

$$A_{mid}(r_{gap}) = \begin{cases} 4\pi R^2 \left(1 \pm \sqrt{1 - \frac{r_{gap}^2}{R^2}}\right), & r_{rim} < r_{gap} < r_{flat} \\ 2\pi r_{flat}^2, & r_{gap} = r_{flat} \end{cases}. \tag{3}$$

Thus, for the flat morphology, where $r_{gap} = r_{flat}$, we can write

$$A = 2\pi r_{flat}^2 + 4\pi r_{rim}\left[h + r_{flat}(\pi - \alpha)\right], \tag{4}$$

which sets the constrained value of the total surface area of a Golgi cisterna. Finally, by using *Equations (2), (3), and (4)* we can mutually relate the two parameters describing cisternae curling, $R$ and $r_{gap}$, as

$$R = \frac{4hr_{rim}(r_{flat}^2 - r_{gap}^2) + r_{flat}^2\left[r_{flat}^2 + 2\pi(r_{flat} - r_{gap})\right] - 2r_{rim}(r_{flat} - r_{gap})r_{flat}^2\alpha}{2r_{flat}\sqrt{(r_{flat} - r_{gap})\left[4hr_{rim}(r_{flat} + r_{gap})r_{flat}^2(r_{gap} + \pi r_{rim} + r_{flat}) + 2r_{rim}r_{flat}^2(\pi/2 - \alpha)\right]}}. \tag{5}$$

In the Appendix we derive simpler version of these equations by imposing a few approximations. This approximate theory will allow us to have analytical estimations of the main numerical results of this article.

## Physical description of the free energy of a Golgi cisterna

Our model considers that the total free energy of a Golgi cisterna, $F$, has two contributions: the first one is the free energy of lipid nanodomain partitioning, $F_{part}$, and the second one is the membrane bending energy, $F_{bend}$.

The free energy of lipid nanodomain partitioning is an entropic term associated with a possible non-homogeneous distribution of liquid-ordered nanodomains along the membrane. This free energy term is given by

$$F_{part} = -TS_{part}, \tag{6}$$

where $T$ is the absolute temperature and $S_{part}$ is the translational entropy associated with the lateral distribution of nanodomains. For the sake of simplicity, we model these domains as small, uniformly sized circular membrane patches of radius $R_d$. The total cisternae membrane area fraction covered by such nanodomains is given by $\Phi = \pi R_d^2 N_d/A$, where $N_d$ is the number of nanodomains. We use the area fraction covered by nanodomains, $\Phi$, as a free parameter in our model. We also have to take

into consideration the dynamics of the SM-enriched nanodomains, which can be continuously formed, reabsorbed and also diffuse along the membrane. However, these processes occur at much faster time scales (of the order of nanoseconds) (*Eggeling et al., 2009*) than the typical time scale of global shape remodeling of the Golgi membranes (of the order of seconds) (*Bankaitis et al., 2012*). Hence, for the purpose of finding how SM levels control the shape of the Golgi membranes, we can disregard domain dynamics and consider that the rigid nanodomains optimally and instantly redistribute along the membrane during cisternae deformation. Taken together these premises, we can write down an expression for the entropy of nanodomain partitioning, using a mean-field approach, as (*Boucrot et al., 2012*; *Kozlov and Helfrich, 1992*; *Shemesh et al., 2003*; *Andelman et al., 1994*; *Derganc, 2007*; *Bozic et al., 2006*)

$$S_{part} = -\frac{k_B}{a_{dom}} \int \left[ \phi\left(\vec{x}\right) ln\phi\left(\vec{x}\right) + \left(1 - \phi\left(\vec{x}\right)\right) ln\left(1 - \phi\left(\vec{x}\right)\right) \right] dA, \qquad (7)$$

where $k_B = 1.38 \times 10^{-23} J \cdot K^{-1}$ is the Boltzmann constant, $a_{dom} = \pi R_d^2$ is the surface area of a single domain, and $\phi\left(\vec{x}\right)$ is the local area fraction of SM nanodomains on the membrane, the area average of which is the total area fraction covered by nanodomains, $\Phi = \frac{1}{A} \int \phi\left(\vec{x}\right) dA$. We consider, for the sake of simplicity, that differential partitioning can only occur between the central region of the cisterna and the cisterna rim, where the membrane curvatures are considerably different from each other. The average nanodomain area fractions in each of these two regions are given by $\Phi_{mid} = \frac{1}{A_{mid}} \int \phi\left(\vec{x}\right) dA$, and $\Phi_{rim} = \frac{1}{A_{rim}} \int \phi\left(\vec{x}\right) dA$, where the integrals are performed over the central and rim areas of the cisterna, respectively. Conservation of the total area fraction covered by nanodomains leads to the following relationship

$$\Phi_{mid} = \Phi + (\Phi - \Phi_{rim})\frac{A_{rim}}{A_{mid}}, \qquad (8)$$

where $A_{rim}$ and $A_{mid}$ are, respectively, the surface area of the rim and central regions of the cisterna (*Equations (2) and (3)*, respectively). Altogether, we can write *Equation (6)* as

$$F_{part} = \frac{k_B T}{\pi R_d^2} [\Phi_{mid} ln\Phi_{mid} + (1 - \Phi_{mid}) ln(1 - \Phi_{mid})] A_{mid} + \frac{k_B T}{\pi R_d^2}$$
$$[\Phi_{rim} ln\Phi_{rim} + (1 - \Phi_{rim}) ln(1 - \Phi_{rim})] A_{rim}, \qquad (9)$$

where $\Phi_{mid}$ is given by *Equation (8)*.

The second contribution to the total membrane free energy comes from the energy of membrane bending, given by the Helfrich Hamiltonian (*Helfrich, 1973*). The membrane bending energy per unit area is given by

$$f_{bend} = \frac{\kappa}{2}(J - J_s)^2 + \bar{\kappa}K, \qquad (10)$$

where $\kappa$ and $\bar{\kappa}$ are the bending modulus and the modulus of Gaussian curvature, respectively; $J$ and $K$ are the total and Gaussian curvatures of the membrane, respectively; and $J_s$ is the spontaneous curvature of the membrane. We assume that a priori there is no spatial correlation between the distribution of rigid nanodomains and of curvature generators. Indeed, SMS-mediated synthesis of SM is restricted to the luminal leaflet of the *trans*-Golgi membranes (*Huitema et al., 2004*), whereas recruitment of curvature generators occurs on the cytosolic side of the membrane. Although transbilayer lipid coupling between phosphatidylserine and long acyl chain lipids has been shown to occur at the level of the plasma membrane (*Raghupathy et al., 2015*), and a specific SM species has been shown to bind transmembrane cargo receptor at the Golgi membranes (*Contreras et al., 2012*), it is not clear how SM-rich nanodomains in the inner leaflet of the *trans*-Golgi membranes are coupled to the budding effector recruitment at the opposed side. The total bending energy of the cisterna is the integral of the free energy density, *Equation (10)*, along the area of the cisterna,

$$F_{bend} = \int f_{bend} dA. \qquad (11)$$

For a laterally inhomogeneous membrane formed by a set of rigid nanodomains, the elastic

moduli vary from regions of high membrane order to regions of low membrane order. We can locally describe the bending modulus of such a membrane as

$$\frac{1}{\kappa\left(\vec{x}\right)} = \frac{\phi\left(\vec{x}\right)}{\kappa_{lo}} + \frac{1 - \phi\left(\vec{x}\right)}{\kappa_{ld}}, \tag{12}$$

where $\kappa\left(\vec{x}\right)$ is the local bending modulus, which has the meaning of an average over soft and rigid membrane domains (*Kozlov and Helfrich, 1992*; *Markin, 1981*). Moreover, $\kappa_{ld}$ and $\kappa_{lo}$ are the bending rigidities of a purely liquid-disordered membrane and of a purely liquid-ordered membrane, respectively. Based on different experimental studies (*Roux et al., 2005*; *Heinrich et al., 2010*), we take the values of these rigidities as $\kappa_{ld} = 20\,k_BT$ and $\kappa_{lo} = 80\,k_BT$. Similarly, $\bar{\kappa}_{ld}$ and $\bar{\kappa}_{lo}$ represent the moduli of Gaussian curvature of purely liquid-disordered and liquid-ordered membranes, respectively. Theoretical considerations, as well as indirect experimental evidence estimate the value of the modulus of Gaussian curvature to be $\bar{\kappa} = \alpha_{\bar{\kappa}}\kappa$, where $\alpha_{\bar{\kappa}}$ is a proportionality factor that ranges between $-0.2$ and $-0.83$ (*Templer et al., 1998*; *Siegel and Kozlov, 2004*). It should be noted that the Gauss-Bonnet theorem cannot be applied to the Gaussian curvature term of the free energy *Equation (11)* since the modulus of Gaussian curvature is not constant along the membrane area (*Allain et al., 2004*). This term is therefore not a topological invariant for laterally inhomogeneous membranes, and therefore needs to be explicitly taken into account. To compute the bending energy in the geometry illustrated in *Figure 1C*, we separately consider the contributions to the elastic free energy of the rim region and of the central region of the cisterna, $F_{bend} = F_{bend}^{mid} + F_{bend}^{rim}$. These two regions are characterized by having an approximately constant total curvature. Hence, as mentioned above, we assume that there is a differential partitioning of SM-rich liquid-ordered nanodomains between the central and the rim regions of the cisternae. The details of the derivation of the expression for the free energy of bending of the cisternae central part, $F_{bend}^{mid}$, and of the cisternae rim, $F_{bend}^{rim}$, are found in the next section.

Finally, the total membrane free energy is given by

$$F = F_{bend}^{mid} + F_{bend}^{rim} + F_{part}, \tag{13}$$

where the individual contributions to the total free energy are given by *Equations (17), (20), and (6)*. In the model we consider that the spontaneous curvature of the membrane could take different values at the rim and central regions, $J_{s,rim}$ and $J_{s,mid}$, respectively. The total membrane free energy *Equation (13)* depends on a set of geometric parameters describing the cisterna morphology, $\{r_{gap}, r_{flat}, r_{rim}, h\}$; a set of nanodomain-related parameters $\{R_d, \Phi, \Phi_{rim}\}$; and a set of parameters describing the elastic properties of the membrane, $\{\kappa_{ld}, \kappa_{lo}, \alpha_{\bar{\kappa}}, J_{s,rim}, J_{s,mid}\}$. A thermodynamic treatment of the curvature effectors could in principle be incorporated into the model. However, in order to reduce the amount of free variables in the model, we distinguished two extreme situations: (i) the membrane bending proteins, contributors to the membrane spontaneous curvature, are only localized at the rims of the Golgi cisternae, implying that $J_{s,mid} = 0$ and $J_{s,rim} = J_s$, which could be explained by the fact that membrane recruitment of some of these proteins is highly sensitive to membrane curvature (*Antonny, 2011*); and (ii) the budding machinery is homogeneously distributed along the whole Golgi membrane, $J_{s,mid} = J_{s,rim} = J_s$. As we showed, the results are qualitatively similar when considering the presence of curvature generators in the central part of the Golgi cisternae, but the shape transition quantitatively shifts. The reason for such a shift comes from the fact that, since the total surface area is conserved, curling of a cisterna leads to an increase in the surface area of the central part of the cisterna, concomitant with a decrease in the surface area of the rim. Hence, increasing $J_{s,mid}$ leads to an increase in the bending energy of this region, thereby penalizing cisterna curling and eventually shifting the flat-to-curled cisterna transition towards smaller values of the spontaneous curvature. In the Appendix we present an analytical estimation of this shift, under certain approximations. In summary, of all the above-mentioned parameters, there are only two free parameters that can change as a result of membrane deformation. The first describes the level of cisternae membrane curling, $r_{gap}$, and the second is associated to the level of nanodomain partitioning between high- and low-curvature membrane regions, $\Phi_{rim}$. Therefore, the optimal gap aperture radius and partitioning of the nanodomains between the rim and the middle part of the cisterna,

$\{r_{gap}^*, \Phi_{rim}^*\}$, correspond to the global minimum of the total free energy in the entire parameter space,

$$F\left(r_{gap}^*, \Phi_{rim}^*\right) = \min_{\forall\{r_{gap}, \Phi_{rim}\}} \left\{F\left(r_{gap}, \Phi_{rim}\right)\right\}. \tag{14}$$

The values of the rest of the parameters are fixed and determined from other studies or vary within a range of possible values (see **Table 1**).

## Mathematical expression of the bending free energy of a membrane cisterna

The derivation of total bending energy of a Golgi cisterna is detailed here, taking separately the contributions from the central part of the cisterna and of the cisterna rim. In the Appendix we also present simplified analytical expressions for the free energy of the cisterna, obtained under certain approximations.

### Central region

The total and Gaussian curvatures along the surface of the central part of the Golgi cisterna are given by $J_{mid} = \pm 2/R$, and $K = 1/R^2$, respectively, where the plus and minus signs in the total curvature value correspond to the bottom and top membrane patches of the central part of the cisterna. Since $h \ll r_{flat}$, the area of these bottom and top membrane surfaces are, to a first approximation, equal, and therefore we can write

$$F_{bend}^{mid} = \frac{\kappa_{mid}}{2}\left(\frac{4}{R^2} + J_{s,mid}^2\right)A + \bar{\kappa}_{mid}\frac{A}{R^2}, \tag{15}$$

where $\kappa_{mid}$ and $\bar{\kappa}_{mid}$ represent, respectively, the bending rigidity and the modulus of Gaussian curvature at the central part of the cisterna; and $J_{s,mid}$ is the spontaneous curvature in the central part of the cisterna. Based on **Equation (12)**, we can write $1/\kappa_{mid} = \Phi_{mid}/\kappa_{lo} + (1 - \Phi_{mid})/\kappa_{ld}$ and $\bar{\kappa}_{mid} = \alpha_{\bar{\kappa}}\kappa_{mid}$. Using these expressions together with **Equations (4), and (8)**, we can rewrite the bending free energy of the middle region **Equation (15)** as

$$F_{bend}^{mid} = \frac{2\kappa_{ld}\kappa_{lo}\pi^2\left[J_{s,mid}^2 \, r_{flat}^2 + 8\left(1 - r_{gap}^2/r_{flat}^2\right)(2 + \alpha_{\bar{\kappa}})\right]}{\kappa_{ld}\left[2\pi\Phi + A_{rim}/r_{flat}^2(\Phi - \Phi_{rim})\right] + \kappa_{lo}\left[2\pi(1 - \Phi) - A_{rim}/r_{flat}^2(\Phi - \Phi_{rim})\right]}. \tag{16}$$

### Rim region

In toroidal coordinates $\{\varphi, \theta\}$ (see **Figure 1—figure supplement 1**), the total and Gaussian curvatures at the rim surface are given, respectively, by

$$J_{rim} = \frac{r_{gap} + 2r_{rim}cos\varphi}{r_{rim}\left(r_{gap} + r_{rim}cos\varphi\right)}, \tag{17}$$

$$K_{rim} = \frac{cos\varphi}{r_{rim}\left(r_{gap} + r_{rim}cos\varphi\right)}. \tag{18}$$

Similarly, the total bending free energy in the rim area is given by

$$F_{bend}^{rim} = \int\int \left[\frac{\kappa_{rim}}{2}\left(J_{rim} - J_{s,rim}\right)^2 + \bar{\kappa}_{rim}K_{rim}\right]dA_{rim}, \tag{19}$$

where $\kappa_{rim}$ and $\bar{\kappa}_{rim}$ are the bending rigidity and the modulus of Gaussian curvature at the cisternae rim area, respectively; and $J_{s,rim}$ is the spontaneous curvature at the rim. This morphology is similar to the one corresponding for fusion pores (**Chizmadzhev et al., 1995**; **Kozlov et al., 1989**). Using **Equation (12)**, we can write $1/\kappa_{rim} = \Phi_{rim}/\kappa_{lo} + (1 - \Phi_{rim})/\kappa_{ld}$ and $\bar{\kappa}_{rim} = \alpha_{\bar{\kappa}}\kappa_{rim}$. The limits of integration in **Equation (19)** when the area element is expressed in toroidal coordinates (**Equation (1)**) are the same as those in **Equation (2)**. This last integral cannot be analytically solved, so we will compute it numerically. In the Appendix, an analytical approximation is obtained under certain simplifying assumptions.

## Strategy of computations

To obtain the local stable shapes of Golgi cisternae as a function of the spontaneous curvature and of the total area fraction of nanodomains, our strategy is to compute the membrane free energy *Equation (13)* as a function of the gap aperture, $r_{gap}$, and the nanodomain area fraction at the cisternae rim, $\Phi_{rim}$. Then, for all values of the gap aperture, $r_{rim} < r_{gap} \leq r_{flat}$, we found the optimal distribution of nanodomains at the cisternae rim, $\Phi_{rim}^*(r_{gap})$, by minimization of the free energy with respect to this parameter,

$$F(r_{gap}, \Phi_{rim}^* r_{gap}) = \min_{\Phi_{rim} \in [0,1]} \{F(r_{gap}, \Phi_{rim}(r_{gap}))\}, \tag{20}$$

*Equation (20)* represents the partitioning-optimized free energy as a function of the gap aperture length (that is, as a function of the cisterna morphology). Depending on the parameter values, the free energy of the cisterna *Equation (20)* has one or two local minima, which correspond to curled or flat cisternae morphologies. Moreover, to compute the energy barriers, we used that $F_{max} = \max_{r_{gap} \in [r_{curl}, r_{flat}]} [F(r_{gap}, \Phi_{rim}^*(r_{gap}))]$, $F_{curl} = F(r_{curl}, \Phi_{rim}^*(r_{curl}))$, and $F_{flat} = F(r_{flat}, \Phi_{rim}^*(r_{flat}))$.

## Free energy including DAG redistribution along the membrane

The free energy of a Golgi cisterna, including the bending energy term taking into account a different distribution of DAG molecules between the top, bottom and rim regions of both the luminal and cytosolic monolayers (see *Figure 5A*), as well as the associated entropic free energy penalty of a non-homogeneous distribution of DAG molecules reads as,

$$\begin{aligned} F = \frac{\kappa}{2} \sum_{i \in \{top,\, bottom, rim\}} \int \left( J_s - \frac{1}{2}\left(\phi_{DAG,cyt}^i - \phi_{DAG,lum}^i\right)\zeta_{DAG} - J_{s,i}\right)^2 dA_i \\ + \frac{k_B T}{a_{DAG}} j \in \{lum, cyt\} \sum \sum_{i \in \{top,\, bottom, rim\}} \left[\phi_{DAG,j}^i ln\phi_{DAG,j}^i + \left(1 - \phi_{DAG,j}^i\right) ln\left(1 - \phi_{DAG,j}^i\right)\right] A_i, \end{aligned} \tag{21}$$

where $J_{s,i}$ are the spontaneous curvatures of the top, bottom, and rim bilayers (index i), $a_{DAG} \approx 0.6\,nm^2$ is the area per DAG molecule (*Shemesh et al., 2003*), and $\phi_{DAG,j}^i$ is the DAG area fraction in the cytosolic or luminal monolayers (index j) of the top, bottom, and rim bilayers (index i). Considering that the total amount of DAG is symmetrically distributed between the luminal and cytosolic monolayers of the Golgi membrane, these quantities are related to the total membrane DAG area fraction, $\phi_{DAG}$, as

$$\sum_{i \in \{top,\, bottom, rim\}} \phi_{DAG,lum}^i A_i = \sum_{i \in \{top,\, bottom, rim\}} \phi_{DAG,cyt}^i A_i = \phi_{DAG} A. \tag{22}$$

## Acknowledgements

We thank members of the Garcia-Parajo lab and Giovanni D'Angelo for valuable discussions. FC and MGP acknowledge support by the Spanish Ministry of Economy and Competitiveness ('Severo Ochoa' Programme for Centres of Excellence in R and D (SEV-2015–240522) and FIS2014-56107-R), BFU2015-73288-JIN, AEI/FEDER;UE, Fundacion Privada Cellex, HFSP (GA RGP0027/2012), EC FP7-NANO-VISTA (GA 288263) and LaserLab 4 Europe (GA 654148). MMK is supported by the Israel Science Foundation. VM acknowledges support from the Spanish Ministry of Economy and Competitiveness through the Programme 'Centro de Excelencia Severo Ochoa 2013–298 2017' (SEV-2012–0208); and support from the CERCA Programme / Generalitat de Catalunya. VM is an Institució Catalana de Recerca i Estudis Avançats (ICREA) professor at the Center for Genomic Regulation and the work in his laboratory is funded by grants from MINECO's Plan Nacional (BFU2013-44188-P), Consolider (CSD2009-00016), and European Research Council (268692). The project has received research funding from the European Union. This paper reflects only the author's views. The Union is not liable for any use that may be made of the information contained therein.

# Additional information

## Competing interests

VM: Senior editor, *eLife*. MMK: Reviewing editor, *eLife*. The other authors declare that no competing interests exist.

## Funding

| Funder | Grant reference number | Author |
|---|---|---|
| Ministerio de Economía y Competitividad | Severo Ochoa Programme SEV-2015-240522 | Felix Campelo<br>María F García-Parajo |
| Fundación Cellex | | Felix Campelo<br>María F García-Parajo |
| Human Frontier Science Program | GA RGP0027/2012 | María F García-Parajo |
| European Commission | FP7-NANO-VISTA GA 288263 | María F García-Parajo |
| Institució Catalana de Recerca i Estudis Avançats | | María F García-Parajo<br>Vivek Malhotra |
| Israel Science Foundation | 758/11 | Michael M Kozlov |
| Ministerio de Economía y Competitividad | Plan Nacional FIS2014-56107-R | María F García-Parajo |
| Ministerio de Economía y Competitividad | Plan Nacional BFU2013-44188-P | Vivek Malhotra |
| Ministerio de Economía y Competitividad | Consolider CSD2009-00016 | Vivek Malhotra |
| H2020 European Research Council | Advanced Grant 268692 | Vivek Malhotra |
| Ministerio de Economía y Competitividad | Severo Ochoa Programme SEV-2012-0208 | Vivek Malhotra |
| Ministerio de Economía y Competitividad | BFU2015-73288-JIN AEI/FED | Felix Campelo<br>María F García-Parajo |

The funders had no role in study design, data collection and interpretation, or the decision to submit the work for publication.

## Author contributions

FC, Conceptualization, Software, Formal analysis, Funding acquisition, Visualization, Methodology, Writing—original draft, Writing—review and editing, Acquisition of data. Computational analysis of the physical model; JvG, Conceptualization, Formal analysis, Visualization, Writing—review and editing, Acquisition of data; GT, SP, Formal analysis, Visualization, Writing—review and editing, Acquisition of data; MMK, Conceptualization, Formal analysis, Supervision, Methodology, Writing—review and editing; MFG-P, VM, Conceptualization, Formal analysis, Supervision, Funding acquisition, Writing—review and editing

## Author ORCIDs

Felix Campelo, http://orcid.org/0000-0002-0786-9548
Vivek Malhotra, http://orcid.org/0000-0001-6198-7943

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

## Appendix 1

In this Appendix, we derive and discuss analytical approximations of the Golgi morphology model, and we compare them to the numerical results presented in the text.

## Approximate analytic expression for the total bending energy of a cisterna

Our first aim is to obtain an analytically treatable, approximate expression for the total bending energy of a cisterna. For this purpose, we start by deriving a simplified expression of the total surface area of the cisterna rim, **Equation (2)**. Assuming that $r_{gap} \gg h$, we get

$$A_{rim} = 4\pi(\pi - \alpha)r_{rim}r_{gap}. \tag{A1}$$

This approximation is most accurate for the case of a flat configuration, where $r_{gap} = r_{flat}$. Using **Equation (A1)** together with **Equation (3)**, we can obtain a simplified version of **Equation (5)**, which describes the relationship between the radius of Golgi curling, $R$, and the length of the cisterna opening, $r_{gap}$,

$$R = \frac{r_{flat}^2 + 2r_{rim}(\pi - \alpha)\left(r_{flat} - r_{gap}\right)}{2\sqrt{\left(r_{flat} - r_{gap}\right)\left(r_{flat} + r_{gap} + 2r_{rim}\right)(\pi - \alpha)}} \approx \frac{r_{flat}}{2\sqrt{1 - r_{gap}^2/r_{flat}^2}} . \tag{A2}$$

Next, we derive approximate expressions for the bending energy of the central and rim regions of a Golgi cisterna, respectively. We consider that the role of the Gaussian curvature is minor, hence we take $\bar{\kappa} = 0$. First, by plugging **Equation (A2)** and **Equation (3)** in **Equation (15)**, we get

$$F_{bend}^{mid} = \kappa_{mid}\pi\left[J_{s,mid}^2 r_{flat}^2 + 16\left(1 - r_{gap}^2/r_{flat}^2\right)\right] . \tag{A3}$$

The following step is to generate an approximate expression for the non-analytical bending energy of the rim region, **Equation (19)**. We assume again a vanishing modulus of the Gaussian curvature, $\bar{\kappa} = 0$. For slightly curled cisternae, we could assume that $r_{gap} \gg r_{rim}$, which would lead to an expression for the total curvature of the rim. **Equation (17)**, $J_{rim} = 1/r_{rim}$. However, this expression does not hold for highly curled cisternae, where $r_{gap} \sim r_{rim}$ (see **Figure 1—figure supplement 1**). In such situation, we can approximate the expression for the total curvature of the rim as

$$J_{rim} = \frac{1}{r_{rim}} - \frac{1}{r'_{gap}}, \tag{A4}$$

where $r_{gap}' = r_{gap} - r_{rim}$. Since this expression introduces only a relatively small error of the order of $r_{rim}/r_{flat} \ll 1$ for almost flat cisternae, we decided to use **Equation (A4)** as a relatively good approximation for the whole range of values of $r_{gap}$. Hence, using **Equations (A1), (A4)**, we can write down an analytical approximation of **Equation (19)** as

$$F_{bend}^{rim} = 2\pi(\pi - \alpha)\kappa_{rim}\frac{\left(r_{gap}' - r_{rim} - r_{gap}' \, r_{rim}J_{s,rim}\right)^2}{r'_{gap} \, r_{rim}} , \tag{A5}$$

which, for relatively flat cisternae, $r_{gap} \gg r_{rim}$, can be further simplified as

$$F_{bend}^{rim}\left(r_{gap} \gg r_{rim}\right) = 2\pi(\pi - \alpha)\kappa_{rim}\frac{\left(1 - J_{s,rim}r_{rim}\right)^2}{r_{rim}}\ r_{gap} = 2\pi\lambda\ r_{gap}, \tag{A6}$$

where we defined

$$\lambda = (\pi - \alpha)\kappa_{rim}\frac{\left(1 - J_{s,rim}r_{rim}\right)^2}{r_{rim}}, \tag{A7}$$

as an effective line tension of the rim, in analogy to the formal description of the lipid bilayer edge energy arising during the formation of small vesicles upon sonication (**Helfrich, 1974**) and of budding membrane domains (**Lipowsky, 1992**). Finally, the total bending energy of the cisterna is the sum of the contributions from the central and rim regions, $F_{bend} = F_{bend}^{mid} + F_{bend}^{rim}$.

## Analytical expression for the cisterna shape transitions

The free energy of nanodomain partitioning, $F_{part}$, has a logarithmic dependence on the nanodomain area fractions, $\Phi_{mid}$ and $\Phi_{rim}$, thus posing a challenge for an analytic treatment (see **Equation 9**). Nevertheless, the degree of nanodomain redistribution only appears in the expression for the total bending energy of a cisterna through the bending rigidities, $\kappa_{rim}$ and $\kappa_{mid}$, given by **Equation (12)**. Hence, to get an analytical approximation of the numerically-computed shape diagram boundaries (see **Figure 2A**), we consider the two extreme situations of nanodomain partitioning, for which we will compute the total bending energy: The first situation corresponds to no nanodomain partitioning, that is to homogeneous distribution of nanodomains along the entire cisterna. From a physical point of view, this situation corresponds to when the total free energy of the cisterna is dominated by the entropic contribution of nanodomain partitioning. The second case corresponds to the complete partitioning of the nanodomains to the central part of the cisterna. From a physical point of view, this corresponds to the bending energy dominating the total free energy of the cisterna. We later discuss about the relative importance of the entropic part of the total free energy of a cisterna as compared to the bending energy.

In both cases, the total free energy of a cisterna reads as

$$F = F_{bend}^{mid} + F_{bend}^{rim}, \tag{A8}$$

where $F_{bend}^{mid}$ and $F_{bend}^{rim}$ are given by **Equations (A3) and (A5)**, respectively. If we further assume that the spontaneous curvature of the central region of the cisterna is zero, we can write **Equation (A8)** as

$$F = 16\pi\ \kappa_{mid}\left(1 - \frac{r_{gap}^2}{r_{flat}^2}\right) + 2\pi(\pi - \alpha)\kappa_{rim}\frac{\left(r_{gap}' - r_{rim} - r_{gap}'\ r_{rim}J_{s,rim}\right)^2}{r_{gap}'\ r_{rim}} \tag{A9}$$

Let us first estimate the critical values of the rim spontaneous curvature that are associated with a loss of stability of flat cisternae, $J_{s,rim}^*$. Mathematically, this corresponds to the situation where the free energy **Equation (A9)** has a local maximum at $r_{gap} = r_{flat}$,

$$\frac{\partial F}{\partial r_{gap}}\bigg|_{r_{gap}=r_{flat},\ J_{s,rim}=J_{s,rim}^*} = 0, \tag{A10}$$

the solution of which leads to the expression for the spontaneous curvature of the rim,

$$J_{s,rim}{}^* \, r_{rim} \simeq 1 - 4\sqrt{\frac{1}{\pi - \alpha}\frac{\kappa_{mid}}{\kappa_{rim}}\frac{r_{rim}}{r_{flat}}}, \qquad (A11)$$

where we assumed that $r_{flat} \gg r_{rim}$. Interestingly, from this expression we can compute the critical effective line tension, **Equation (A7)**, as $\lambda^* = \lambda\left(J_{s,rim}^*\right) = 8\,\kappa_{eff}/r_{flat}$, where $\kappa_{eff} = 2\kappa_{mid}$ is an effective total bending rigidity of the central part of the cisterna, given that it is made of two parallel bilayers. This rough estimation matches with previously calculated analytical values for critical line tensions required to form spherical vesicles by sonication (**Helfrich, 1974**) or budded membrane domains (**Lipowsky, 1992**).

We next estimate the critical value of the rim spontaneous curvature associated with a loss of stability of a highly curled cisterna, $J_{s,rim}^{**}$. Basically, this condition corresponds to a situation where the total free energy **Equation (A9)** ceases to have a local minimum for a highly curled cisterna configuration. The local extremes of the free energy function are given by the solutions of the equation $\frac{\partial F}{\partial r_{gap}} = 0$, which can be approximated as

$$-16\,\frac{\kappa_{mid}}{\kappa_{rim}}\frac{1}{r_{flat}^2}r_{gap}{}^3 + \frac{\pi - \alpha}{r_{rim}}\left(1 - r_{rim}J_{s,rim}\right)^2 r_{gap}{}^2 - (\pi - \alpha)r_{rim} = 0 \qquad (A12)$$

which is a cubic equation in $r_{gap}$, with coefficients $a = -16\,\frac{\kappa_{mid}}{\kappa_{rim}}\frac{1}{r_{flat}^2}$, $b = \frac{\pi-\alpha}{r_{rim}}\left(1 - r_{rim}J_{s,rim}\right)^2$, $c = 0$, and $d = -(\pi - \alpha)r_{rim}$. Loss of a local minimum for a highly curled configuration, mathematically, corresponds to **Equation (A12)** having only one real solution (and two complex conjugate solutions) instead of three real solutions. The transition in the number of real solution of a cubic equation corresponds to when the discriminant of the cubic equation, $\Delta = -4b^3 d - 27a^2 d^2$, is equal to zero (see, for instance, [**Abramowitz and Stegun, 1964**]). This condition, after some algebra, leads to

$$J_{s,rim}{}^{**}\, r_{rim} = 1 - 2\sqrt{3}\left(\frac{1}{\pi - \alpha}\frac{\kappa_{mid}}{\kappa_{rim}}\frac{r_{rim}{}^2}{r_{flat}{}^2}\right)^{1/3}. \qquad (A13)$$

We can now estimate the values of the transition spontaneous curvatures, **Equations (A11) and (A13)**, which correspond to the aforementioned extreme situations of complete partitioning and homogeneous distribution of nanodomains.

*Homogeneous nanodomain distribution:* This situation corresponds to the condition $\Phi_{rim} = \Phi_{mid} = \Phi$, and therefore $\kappa_{rim} = \kappa_{mid} = \kappa$. Under these circumstances, it is straightforward to see that the transition zones do not depend on the area fraction covered by nanodomains, $\Phi$, or on the value of the bending rigidity of the membrane, $\kappa$, which is the only energy scale of the system. Hence, **Equations (A11) and (A13)** can be, respectively, expressed as

$$J_{s,rim}{}^{*,hom}\, r_{rim} = 1 - 4\sqrt{\frac{1}{\pi - \alpha}\frac{r_{rim}}{r_{flat}}}, \qquad (A14)$$

and

$$J_{s,rim}{}^{**,hom}\, r_{rim} = 1 - 2\sqrt{3}\left(\frac{1}{\pi - \alpha}\frac{r_{rim}^2}{r_{flat}^2}\right)^{1/3}. \qquad (A15)$$

*Complete nanodomain partitioning:* As explained above, this situation corresponds to the condition $\Phi_{rim} = 0$, and $\Phi_{mid} \simeq \Phi$. According to **Equation (12)**, the bending rigidities at the

rim and central part of the cisterna are expressed as $\kappa_{rim} = \kappa_{ld}$ and $\kappa_{mid} = \kappa_{ld}/(1 - \Phi(1 - \kappa_{ld}/\kappa_{lo}))$. Now, we can rewrite **Equations (A11) and (A13)** as

$$J_{s,rim}{}^{*,part} \, r_{rim} = 1 - 4\sqrt{\frac{1}{\pi - \alpha}\frac{1}{1 - \Phi(1 - \kappa_{ld}/\kappa_{lo})}\frac{r_{rim}}{r_{flat}}}, \tag{A16}$$

and

$$J_{s,rim}{}^{**,part} \, r_{rim} = 1 - 2\sqrt{3}\left(\frac{1}{\pi - \alpha}\frac{1}{1 - \Phi(1 - \kappa_{ld}/\kappa_{lo})}\frac{r_{rim}{}^2}{r_{flat}{}^2}\right)^{1/3}. \tag{A17}$$

In **Appendix 1—figure 1**, we plot these analytical approximations of the transition zones of the shape diagram, **Equations (A14–A17)**, and compare them to the numerical computations shown in **Figure 2A**. These results show that our analytical approximation is good enough to qualitatively estimate the shape diagram of flat-to-curled cisterna transitions, as well as the shape bistability. However, these results differ quantitatively from the numerical solutions by about 20% (see **Appendix 1—figure 1**).

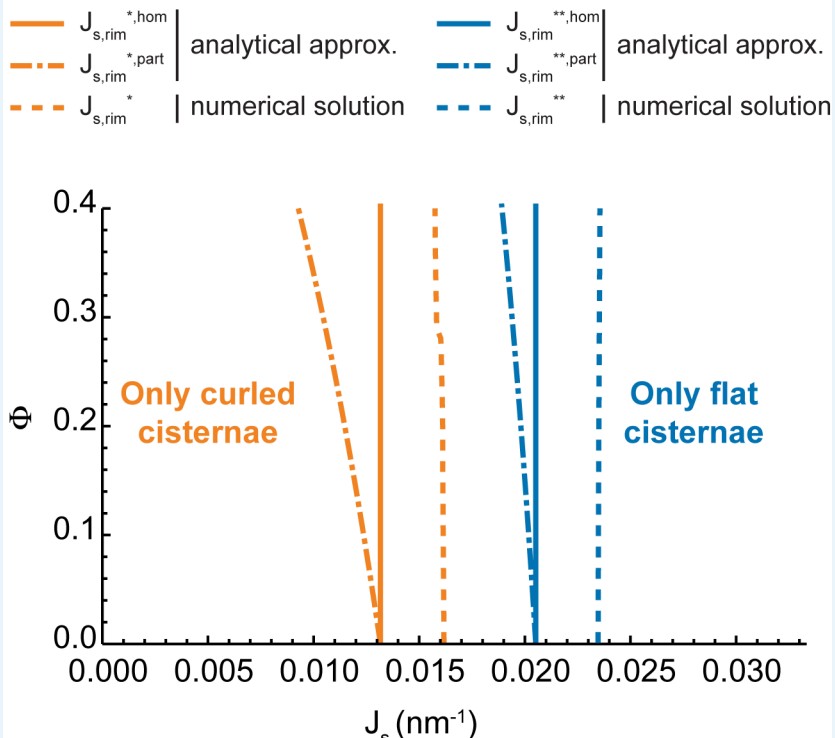

**Appendix 1—figure 1.** Analytical approximation of the shape diagram. Comparison between the numerically-computed shape diagram of a Golgi cisterna (simple dashed lines) to the one obtained by using an analytical approximation, for homogeneous nanodomain distribution (solid lines) and complete nanodomain partitioning (dot-dashed lines). We used the same elastic and geometric parameters as in **Figure 2A** (see **Table 1**). Orange lines represent the transition lines below which only curled cisternae are locally stable shapes, whereas blue lines represent the loss of local stability of curled cisternae.

# Analytical description of the relative contribution of the entropic partitioning free energy to the total free energy of the system

Our numerical results showed that the contribution of the nanodomain area fraction to the Golgi cisterna shape diagram is relatively minor (**Figure 2A**). Moreover, increasing the size of the nanodomains increased the sensitivity of the shape transitions to the amount of nanodomains (**Figure 4**). To gain some more insight into the role of the nanodomain area fraction on the Golgi cisterna shape transition, we aimed at understanding the relative contribution to the overall cisterna free energy of the entropic free energy of nanodomain partitioning and the bending energy. For the sake of simplicity, we focus on the flat cisterna configuration, corresponding to $r_{gap} = r_{flat}$. In this case, and assuming that $r_{rim} \ll r_{flat}$, we can express the bending energy of the cisterna, **Equation (A9)**, as

$$F_{bend} = 2\pi(\pi - \alpha)\kappa_{rim}\frac{r_{flat}}{r_{rim}}\left(1 - r_{rim}J_{s,rim}\right)^2 \tag{A18}$$

where, according to **Equation (12)**,

$$\kappa_{rim} = \frac{\kappa_{lo}\kappa_{ld}}{\kappa_{lo} - (\kappa_{lo} - \kappa_{ld})\Phi_{rim}} = \frac{\kappa_{lo}\kappa_{ld}}{\kappa_{lo} - (\kappa_{lo} - \kappa_{ld})\Phi + (\kappa_{lo} - \kappa_{ld})\Phi\xi}, \tag{A19}$$

where $\xi = (\Phi - \Phi_{rim})/\Phi$ is the nanodomain partitioning coefficient, which ranges between $\xi = 0$ for a non-partitioned homogeneous nanodomain distribution to $\xi = 1$ for fully partitioning of nanodomains away from the cisterna rim. Noteworthy, the bending energy **Equation (A18)** is a monotonically decreasing function for $0 < \xi < 1$, as long as $\kappa_{lo} > \kappa_{ld}$. Also assuming that $r_{rim} \ll r_{flat}$, we can express the partitioning free energy, **Equation (9)**, as

$$F_{part} = \frac{2k_BT\,r_{flat}^2}{R_d^2}\left([\Phi ln\Phi + (1 - \Phi)ln(1 - \Phi)] + 2(\pi - \alpha)\frac{r_{rim}}{r_{flat}}[\Phi_{rim}ln\Phi_{rim}\right.$$
$$\left. + (1 - \Phi_{rim})ln(1 - \Phi_{rim})]\right), \tag{A20}$$

where $\Phi_{rim} = \Phi(1 - \xi)$. In this case, the partitioning free energy, **Equation (A20)** is a monotonically increasing function for $0 < \xi < 1$, as long as $\Phi < 1/2$. Given this opposite monotonicity between the bending energy, **Equation (A18)** and the partitioning free energy, **Equation (A20)**, the relative contribution of these two energies will dictate whether the partitioning entropy is the dominant part of the free energy, and therefore the nanodomains will tend to be homogeneously distributed along the cisterna; or the bending energy dominates and an extensive partitioning is to be expected. To have a rough analytical estimation of this relative contribution, we look for the condition where the growth slopes of the two energies cancel each other out when there is no nanodomain partitioning, that is

$$\frac{\partial\left(F_{bend} + F_{part}\right)}{\partial\xi}\Big|_{\xi=0} = 0, \tag{A21}$$

which, after some algebra, leads to the expression of a critical nanodomain size, $R_d^*$,

$$R_d^* = \frac{2\,r_{rim}}{\sqrt{\pi}}\frac{1 - (1 - \kappa_{ld}/\kappa_{lo})\Phi}{\sqrt{\kappa_{ld}(1 - \kappa_{ld}/\kappa_{lo})/k_BT}}\frac{\sqrt{\tanh^{-1}(1 - 2\Phi)}}{|1 - r_{rim}J_{s,rim}|}, \tag{A22}$$

below which entropy dominates and low nanodomain partitioning is expected; and above which bending energy dominates and nanodomains are mostly distributed away from the cisterna rim to minimize the overall free energy. In *Appendix 1—figure 2*, we plot the value of the critical nanodomain size as a function of the spontaneous curvature of the rim and/or the area fraction covered by nanodomains. These results indicate that, indeed, the free energy penalty of nanodomain redistribution dominates over the bending energy for small values of the nanadomain size, and therefore large partitioning away from the rim is entropically prevented.

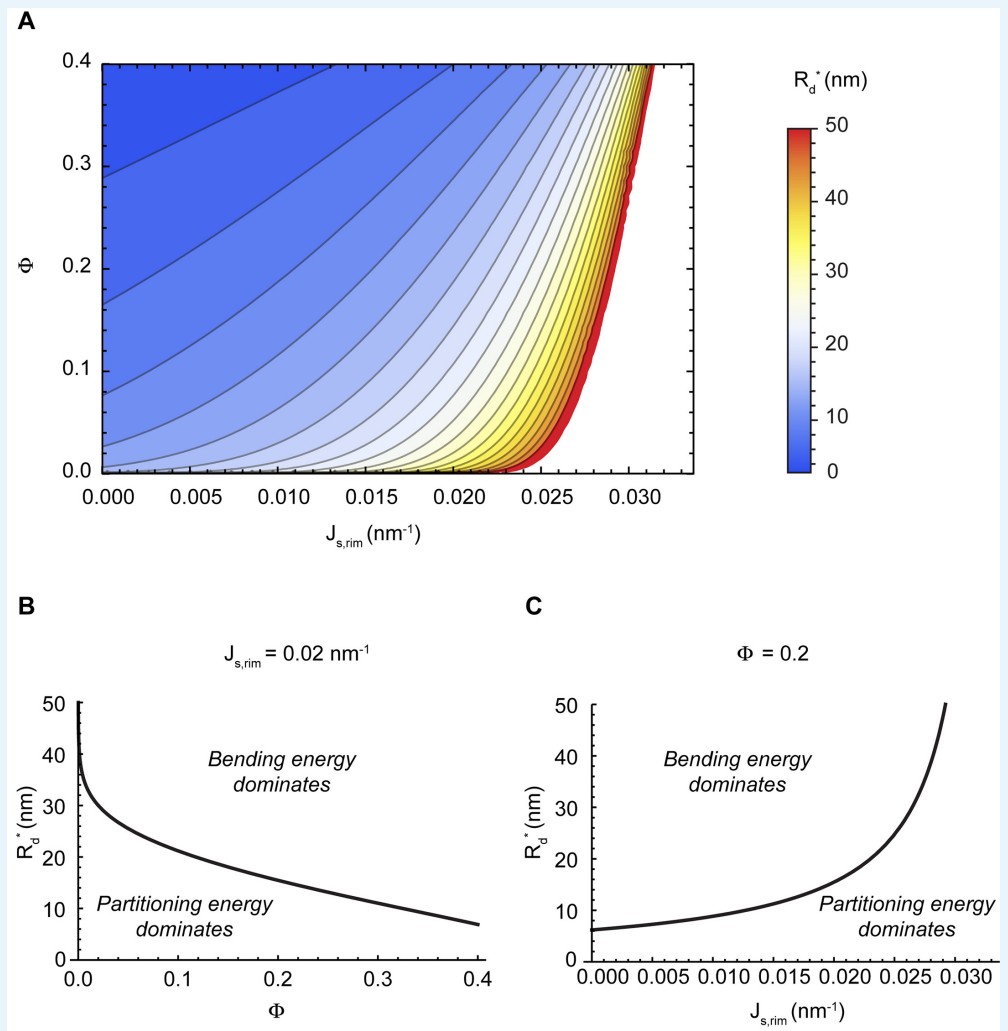

**Appendix 1—figure 2.** Comparison of the partitioning and bending energy terms of the free energy. Critical nanodomain size as a function of the nanodomain area fraction and the spontaneous curvature of the rim (**A**), or plotted separately as a function of the nanodomain area fraction for a fixed value of the rim spontaneous curvature (**B**), and as a function of the rim spontaneous curvature for a fixed nanodomain area fraction (**C**). In (**B**) and (**C**), the regions where bending energy or partitioning energy dominate are indicated.

# Analytical estimation of the role of the distribution of membrane curvature generators in the cisterna shape diagram

In *Figure 2—figure supplement 1*, we numerically showed that the presence of curvature generators on the central part of the Golgi cisterna does not qualitatively change the shape diagram but shifts the flat-to-curled transitions to lower values of the spontaneous curvature. To better understand this change and have a rough analytical estimation of the extent of the shift, we proceeded to compare the total bending energy (assuming that there are no rigid nanodomains) of a flat cisterna to a simplified situation of a completely curled cisterna without a rim (that is, corresponding to two concentric spheres, separated by a distance $2h$ and with a mean radius $R$). In the flat configuration, the surface area of the rim is $A_{rim}^{flat} = 4\pi(\pi - \alpha)r_{rim}r_{flat}$, while the surface area of the central part is $A_{mid}^{flat} = 2\pi\, r_{flat}^2$, and the total surface area of the cisterna is $A = A_{rim}^{flat} + A_{mid}^{flat}$. In the completely curled configuration, there is no rim, so the total surface area is $A = A_{mid}^{curled} = 8\pi\, R^2$. Surface area conservation leads to $R = \frac{1}{2}\, r_{flat}\sqrt{1 + 2(\pi - \alpha)r_{rim}/r_{flat}}$. The bending energies for the rim and central parts of the flat cisterna are, respectively,

$$F_{rim}^{flat} = \frac{\kappa}{2}\left(\frac{1}{r_{rim}} - J_{s,rim}\right)^2 A_{rim}^{flat}, \tag{A23}$$

$$F_{mid}^{flat} = \frac{\kappa}{2}\left(J_{s,mid}\right)^2 A_{mid}^{flat}. \tag{A24}$$

The bending energy of the completed curled cisterna made of two concentric spheres with approximately opposite total curvature $\pm 1/R$ is

$$F_{mid}^{curled} = \frac{\kappa}{2}\left(\frac{4}{R^2} - J_{s,mid}^2\right)A_{mid}^{curled}. \tag{A25}$$

The condition where the two shapes have the same total bending energy can be mathematically expressed as $F_{rim}^{flat} + F_{mid}^{flat} = F_{mid}^{curled}$, characterized by a rim spontaneous curvature $J_{s,rim}^{(0)}$. Plugging all these expressions together, we obtain the following expression,

$$\left(1 - J_{s,rim}^{(0)}r_{rim}\right)^2 - J_{s,mid}^2 r_{rim}^2 = \frac{8}{(\pi - \alpha)}\frac{r_{rim}}{r_{flat}}. \tag{A26}$$

When the spontaneous curvature of the central part is zero, $J_{s,mid} = 0$, we obtain

$$J_{s,rim}^{(0)}\left(J_{s,mid} = 0\right) = \frac{1}{r_{rim}} - 2\sqrt{\frac{2}{(\pi - \alpha)r_{rim}r_{flat}}}. \tag{A27}$$

Alternatively, when the spontaneous curvature generators are all over the cisterna membrane, $J_{s,mid} = J_{s,rim}$, we obtain

$$J_{s,rim}^{(0)}\left(J_{s,mid} = J_{s,rim}\right) = \frac{1}{2\,r_{rim}} - \frac{4}{(\pi - \alpha)r_{flat}}. \tag{A28}$$

We can then compute the relative change in the critical spontaneous curvature as

$$\delta J_{s,rim}^{(0)} = \frac{J_{s,rim}^{(0)}\left(J_{s,mid} = J_{s,rim}\right) - J_{s,rim}^{(0)}\left(J_{s,mid} = 0\right)}{J_{s,rim}^{(0)}\left(J_{s,mid} = 0\right)} = \frac{1}{2} - \sqrt{\frac{2}{(\pi - \alpha)}\frac{r_{rim}}{r_{flat}}}. \tag{A29}$$

For the typical parameters we used in our computations (e.g. in **Figure 2—figure supplement 1**), we obtain $J_{s,rim}^{(0)}\left(J_{s,mid} = 0\right) = 0.0191 \; nm^{-1}$, $J_{s,rim}^{(0)}\left(J_{s,mid} = J_{s,rim}\right) = 0.0136 \; nm^{-1}$, and therefore $\delta J_{s,rim}^{(0)} = -0.29$, which compare remarkably well with the numerically calculated shift in the shape transition (**Figure 2—figure supplement 1**).

## Analytical considerations of the effect of multiple stacked cisternae on the shape diagram

We now consider the effect of multiple stacked cisternae on the flat-to-curled and curled-to-flat cisterna shape transitions. Let us consider a system composed of $N$ cisternae, which are connected along the stack by certain stacking factors. Derganc et al. presented a mechanical model of the Golgi stack, comparing the bending energy of each cisterna with the adhesion energy keeping them together, aiming to understand the physical mechanisms setting the number of cisternae in a flat stack (**Derganc et al., 2006**). Since in the course of our experiments we did not observe major unstacking of the Golgi cisternae, we assume that the adhesion energy keeping the cisternae stacked together is much stronger than the rest of energies (namely, the bending energy), implying that all cisternae curl equally. Moreover, since the lateral size of a cisterna is much larger than the thickness of the stack, we take the approximation that all cisternae acquire the same curvature upon curling. For the sake of simplicity, we disregard the effects of nanodomain partitioning along the membrane (that is, we assume there is no nanodomain partitioning $\Phi_{rim} = \Phi_{mid} = \Phi$). In addition, we need to ascribe the distribution of the spontaneous curvature along the different cisternae of the stack. Let us consider a situation where the stack is composed of $t$ *trans*-like cisternae with a characteristic rim spontaneous curvature, $J_{s,rim}{}^{trans}$, and $N - t$ *cis/medial*-like cisternae, characterized by a different rim spontaneous curvature, $J_{s,rim}{}^{cis}$. The total free energy of the stack is given by the sum of the free energies of each cisterna,

$$F = \sum_{i=1}^{N} F_i = t \, F_{trans} + (N - t) F_{cis}, \tag{A30}$$

where $F_{trans}$ and $F_{cis}$ are given by **Equation (A9)**, using $J_{s,rim} = J_{s,rim}{}^{trans}$ and $J_{s,rim} = J_{s,rim}{}^{cis}$, respectively, and $\kappa_{rim} = \kappa_{mid} = \kappa$. Similarly as we did for a single cisterna, we can obtain analytical expressions for the regions of loss of stability of both flat and curled stacks, given by

$$J_{s,rim}{}^{trans,*}\left(J_{s,rim}{}^{cis}\right) r_{rim} \simeq 1 - 4\sqrt{\frac{N}{t}\frac{1}{(\pi - \alpha)}\frac{r_{rim}}{r_{flat}} - \frac{1}{16}\frac{(N - t)}{t}\left(1 - J_{s,rim}{}^{cis} r_{rim}\right)^2}, \tag{A31}$$

and

$$J_{s,rim}{}^{trans,**}\left(J_{s,rim}{}^{cis}\right) r_{rim} \simeq 1 - \sqrt{12\frac{N}{t}\left(\frac{1}{(\pi - \alpha)}\frac{r_{rim}^2}{r_{flat}^2}\right)^{2/3} - \frac{N - t}{t}\left(1 - J_{s,rim}{}^{cis} r_{rim}\right)^2}, \tag{A32}$$

respectively. It is easy to see that **Equations (A31) and (A32)** correspond to **Equations (A11) and (A13)**, respectively, in the limit $t = N$, that is, when all cisternae are *trans*-like. In **Appendix 1—figure 3** we plotted these transition curves for a stack composed

of 3 *cis*-like cisternae and 1 *trans*-like cisterna, and compared them to the single *trans*-like cisterna situation. These results indicate that curling of the whole stack requires changes in the spontaneous curvatures of all the cisternae. Moreover, we can see that when the two spontaneous curvatures change in the same manner, the shape transitions are independent on the number of cisternae and correspond to the values obtained for a single cisterna (see *Appendix 1—figure 3*).

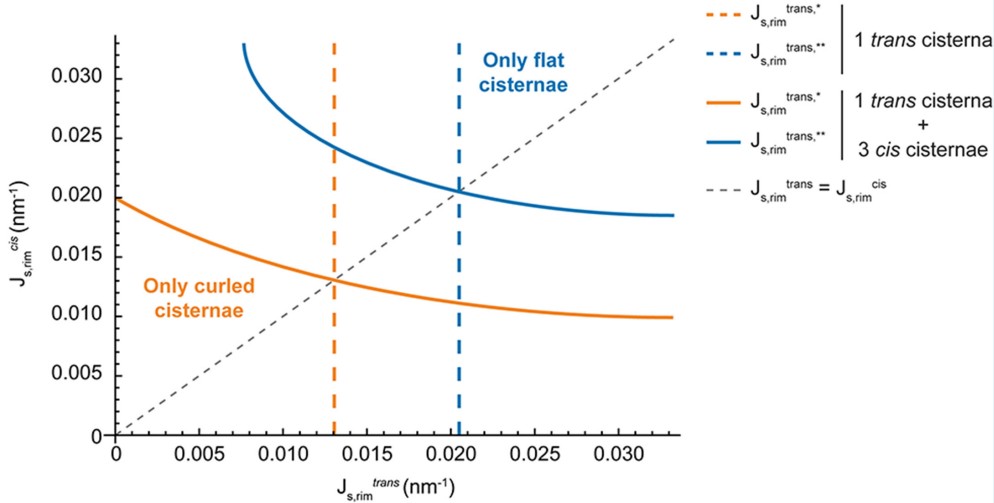

**Appendix 1—figure 3.** Effect of multiple stacked cisternae on the Golgi shape diagram. The transition curves where flat (orange) or curled (blue) cisternae cease to be locally stable configurations are shown for a stack composed of 1 *trans*-like cisterna and 3 *cis*-like cisternae (solid lines) and compared to the situation of a single *trans*-like cisterna (dashed lines). In addition, we show the path where the spontaneous curvatures of the *trans*-like and *cis*-like cisternae are equal (grey dashed line), showing that under this constraint, the shape diagram does not depend on the number of cisternae of each kind.

