## [Decision Letter]

Thank you for submitting your article "Sphingomyelin metabolism controls the shape and function of the Golgi cisternae" for consideration by *eLife*. Your article has been favorably evaluated by Jonathan Cooper (Senior Editor) and three reviewers, one of whom is a member of our Board of Reviewing Editors. The following individual involved in review of your submission has agreed to reveal his identity: Pierre Sens (Reviewer #2).

The reviewers have discussed the reviews with one another and the Reviewing Editor has drafted this decision to help you prepare a revised submission.

All three reviewers appreciated very much your effort to compare your theoretical model on Golgi shape and its bistability to experimental data. However, some concerns were raised on hypotheses that underlie your model such as an equilibrium statement or the absence of discussion about the consequence of lipid fluxes. Moreover, the referees found that your experimental data should be reinforced to convincingly support your model. You will see that there is good agreement between the reviewers. Notably they both have difficulty in reconciling the following three points (i) your model ignores the cis/trans asymmetry of the Golgi, (ii) your experimental approach seems to target only the coating of trans side, (iii) however, all cisternae similarly respond to this treatment.

At the risk of burdening you with excessive feedback, we have appended a somewhat truncated version of the reviews, to help you identifying points to be clarified. While this is not the typical *eLife* style, we think the high quality of the feedback justifies giving it to you in largely unedited form.

*Reviewer #1:*

1) The main assumption of the model is that "The equilibrium configuration of a cisterna is assumed to correspond to the free energy minimum." The author should explain the extent to which free energy minimisation is meaningful for such non-equilibrium structure as the Golgi apparatus, which is characterised by fluxes of proteins and lipids.

2) The modelling strategy and results, in particular the existence of bistability, are not particularly new for membrane systems. Nevertheless, its application to Golgi cisternae shapes is to my knowledge new. The model and the derivation of the results are well explained. I find it however very unfortunate that the results are mostly presented as numerical results, varying a handful of the many model parameters. It would be much more attractive to have analytical expression for the boundaries of the phase diagram. The full model can obviously not be analytically solved, but the main concepts can be included in a much simpler, and solvable model. The essence of the model is a bendable circular domain with edges. This is essentially the same situation as a membrane domain with a line tension, as is discussed by R. Lipowsky (J. Phys. 1992, 2,1825) and many others. This model also predicts bistability (Figure 3 in Lipowsky's paper) where flat and invaginated domains can coexist. This reference should be cited and discussed.

3) The physical mechanism underlying this paper is not new, but its application to the shape of closed membranes (vesicles, cisternae) is new, to my knowledge. There is however a vast literature on the shape of vesicles, including stomatocyte shape not dissimilar to the curled cisternae, but driven by completely different mechanisms (such as area to volume ratio variation, see for instance U. Seifert Adv Phys. 1997, 46, 1 – Figure 14). This is not discussed at all in this paper, which is rather unfortunate. This paper should make clear that it proposes one possible scenario to explain the observations, but that other scenarii exists.

4) Furthermore, previous work describing energetic description of Golgi cisternae, including phase separation of membrane components (Derganc et al. Traffic 2006; 7: 85-96, Derganc Phys. Biol. 4 (2007) 317-324), should definitely be cited and the present results compared to the existing one when applicable, in particular concerning the curvature-coupled lateral segregation.

5) The theory part of the Materials and methods section gives a full geometrical description of the cisterna morphology, yielding rather cumbersome expressions (for ex. Equation.5). While these expressions are probably useful, approximate expressions allowing to derive the transition boundaries in Figure 2 and others would be more useful. A large part of the paper is devoted to explaining the effect of varying some parameters while keeping other fixed. This a rather arbitrary numerical exercise. For instance, the phase diagram of Figure 2 shows the effect of the spontaneous curvature (1/nm), while this parameter only appears in the energy coupled to one or several length scales (certainly the rim radius, and possibly the size of the flat region). Working out analytical expression with some simplifying assumptions would provide much more insights. Using the approach of a circular domain with line tension (dependent on the bending rigidity and the geometry of the rim), and neglecting the entropy of the nanodomains, it is quite easy to find analytical expression for the loss of stability of the flat cisterna. This shows that if the bending rigidity of the rim and the flat part are the same, this parameter drops out and the threshold spontaneous curvature times the rim radius has a simple expression in terms of the ratio of rim radius to cisternae radius. Such analysis should be performed and compared to the numerical results.

6) Such a simplified analytical model would allow to tease apart the different contributions of the overall free energy. It is reported (subsection “Prediction of flat and curled cisterna configurations”, sixth paragraph) that the fraction of rigid domains has little impact on the transition. This observation can mean two things: either the additional bending rigidity associated to the nanodomains is irrelevant, or it is compensated by the entropy cost of partitioning. In the absence of partitioning, one can easily understand that the bending rigidity plays no role in the transition, since it is the only energy scale in the problem, as discussed above. The effect of partitioning should be assessed by comparing entropic and elastic terms analytically (even if that means using approximate expressions).

7) There is a number of simplifying assumptions in the model that do not appear justified and might well influence the results.

7A) The most serious is, to my mind the fact that the radius of the rim is not optimised, but imposed arbitrarily. I can accept the fact that the luminal space h can be fixed by spacer proteins, the rim radius is probably fixed by some minimisation of elastic energy, and should highly depend upon the rim spontaneous curvature. Do the authors have argument to validate their assumption of rim radius independent of the rim spontaneous curvature? I would be looking for theoretical arguments, not experimental argument based on ultrastructural data. Since an energetic description is likely to suggest that the rim radius should depend on its spontaneous curvature, if it is observed experimentally that it doesn't, this would mean that some energetic contributions are not included in the current energy, thus casting doubt on the consistency of the theory with the data.

7B) We are dealing here with a model of a mixed membrane containing at least three different types of molecule: some "neutral" lipids, rigid nanodomains, and curvature generators. It is assumed that such mixed, three component membrane can still be described by one bending rigidity one gaussian rigidity, and one spontaneous curvature. The expression of the composite bending rigidity (Equation 12) corresponds to heterogeneous curvature distribution (i.e., the stiffer regions are less curved than the softer ones, for a given mean curvature). It is thus very likely that the total curvature energy of this mixed membrane with curvature generators will be very different from the one proposed here if the curvature generators bind preferentially to one of the two components. Apparently, SM regulates both membrane stiffness and the binding of curvature generators, so spontaneous curvature and bending rigidity are probably spatially correlated. The authors should discuss this point.

8) Related to 7B): The Abstract claims to investigate the dual effect of SM, which is to affect the rigidity of membranes by forming stiff domains, and to regulates the association of curvature-generating proteins. It is not clear to me what whether SM promotes or inhibit the recruitment of curvature generators. Is this known/implemented in the model?

In the model, the former is associated to a fraction *Φ_budding_* and the latter to a fraction *Φ*, but no connection is made between these two fractions. If SM mediate both effects, should *Φ_budding_* and *Φ* be connected?

Why is the partitioning of the nano-domain partition between the rim and the mid-section subjected to a thermodynamic treatment (enthalpy vs. entropy), but not the curvature agents? This seems a rather arbitrary choice.

9) When the entire cisterna including the mid part, has spontaneous curvature, one observes "only a general shift of the shape transitions between the flat and curled cisterna configurations towards the lower values of the membrane spontaneous curvature". It is not clear to me why this shift occurs, since if both sides of the cisterna have the same spontaneous curvature, its contribution to the global energy can only depend on the area difference between the convex and concave sides of the cisterna (which have curvatures of opposite sign). I would then expect the contribution of the spontaneous curvature of the mid part to the energy difference between flat and curled to be or order Κ J_s_ h~ Κ/2. This seems quite small compared to the overall energy scale, and does not explain the rather large shift observe Figure 1—figure supplement 2. Could the author provide an estimate for the effect of the spontaneous curvature of the flat part.

10) Related to this, the shape parametrisation is clear for flat cisternae, but a little less clear for curled one (Figure 1). My understanding is that since the total are is conserved, and the area of the rim decreases upon curling, the area of the mid part must increase for curled cisternae. This increase is distributed in some way between the concave and convex sides of the cisterna. I would not expect these details to matter much, but since there is an effect of the spontaneous curvature of the mid part, that I can only understand based on the area difference between the convex and concave regions, the details might matter. Judging from Figure 1, both mid regions are defined by the same opening angle, and are hence characterised by a relative area difference Δ S/S ~ h/R. There is no real reason for this to be the case, and there is more leeway here for energy minimisation depending of the spontaneous curvature. Could the authors comment on this? If the conclusion is that these details do matter, it speaks against the general validity of the proposed energetic description.

11) The effect above is discussed: "the results are highly insensitive to the amount of curvature generators present in the central part of the Golgi cisternae, basically due to the opposite curvature of the two parallel membranes composing this region." However, Figure 2—figure supplement 1 shows a shift of about 50% of the transition boundaries. Since this paper is a practical application of well-known concepts, rather than the description of new concepts, I guess numerical values are important and the author should provide an estimate of this shift using simple analytical models.

12) The analysis of drug treatment and recovery is the light of the bistability discussed in this paper is quite interesting. I however do not fully understand the implication of the loss, and the recovery, of TGN46 and p230 colocalisation. These proteins appear to be TGN proteins, while this paper discusses the morphology of Golgi cisternae. Furthermore, I thought that clathrin was also mostly localised at the TGN and was responsible for transport between the TGN and the endosomes/plasma membrane. Is the curling of the Golgi cisternae accompanied with structural changes of the TGN? Should we understand the Golgi and the TGN as a single structure? The concentric stacked onion-like structure presumably involves curling of all Golgi cisternae. Where is the TGN located? In the convex or concave region of the curled stack?

13) Along the same lines: the curling phenomenon is a spontaneous symmetry breaking, which predict an equal likelihood to curl one way or the other. Is curling equally likely in both direction (i.e. toward or away from the TGN) in experiments? If there is a net tendency to curl one way, that means that symmetry is broken from the start by something. This could be because the line tension effect mostly impact the TGN, which then drive the entire Golgi to curl, but in this case, the energy function should include the other Golgi cisternae. It could also be that symmetry is broken by anterograde or retrograde fluxes, in which case the entire model is problematic. More discussion on this is mandatory.

14) In the mind of the authors, is the lateral segregation discussed upon D-cer-C6 treatment related to a lateral partitioning such as the one quantified in Figure 3? In any case, the observation that colocalisation is back to untreated level after 16h in a 30' treatment (Figure 6), while about half the Golgi stacks still show a curled shape (Figure 6) seems to suggest that partitioning and curling are not directly related, unlike the author claim.

15) The level of clathrin in the Golgi reaches its lowest level after about 30'. This suggest that the loss of the curvature generators saturates after that exposure time (Figure 6). However, the recovery of colocalisation is much faster (12h) after 30' treatment than after 4h treatment (12h vs. 24h). This suggest that the recovery time depends on the treatment duration, while the bistability picture would suggest that it only depends on the energy barrier, which is related to the clathrin level, and should thus saturate after 30'. Can the author comment on this, which in my mind argues against the bistability picture?

*Reviewer #2:*

Theoretical part

1) The authors write: 'The equilibrium configuration of a cisterna is assumed to correspond to the free energy minimum'. Given the emphasis put in local energy minima later in the text, this statement is a little bit confusing.

2) The authors write: 'the results, presented in Figure 2—figure supplement 2, show that both the energy barrier required to flatten a curled cisterna, Δ𝐹𝑐𝑢𝑟𝑙−𝑓𝑙𝑎𝑡, and the energy barrier required to curl a flat cisterna, Δ𝐹𝑓𝑙𝑎𝑡−𝑐𝑢𝑟𝑙, decrease with the amount of rigid nanodomains on the Golgi membranes. It seems to me that what is shown here is rather an increase.

3) In a previous study, the authors proposed an alternative mechanism for the flat to curled shift: “We suggest that these changes are a consequence of the effect of d‐ceramide‐C6 on membrane fission versus membrane fusion at the Golgi membranes. In the presence of d‐ceramide‐C6, transport carriers fuse with the Golgi membranes, however, membrane export is blocked. This increases the surface area‐to‐volume ratio of the Golgi cisternae. This is similar to morphological changes in model membranes as a result of an increase in the surface area‐to‐volume ratio.” Why is this hypothesis no longer considered here?

4) A few theoretical papers on the discoïdal shape of membranes or Golgi stacks exist, notably by Derganc, Lipowski, and Svetana. Please comment, compare…

5) In terms of curvature, is the curled shape so highly curved as stated in the Abstract? (…affecting SM homeostasis converted flat cisternae into highly curved membranes). Looking at the EM, the curvature of the cup shape cisternae seems rather modest.

For the experimental part I have the following concerns:

6) The lipid composition and the nature of protein coats associated with the cis and trans faces of the Golgi apparatus are very different. Therefore, why does the Golgi apparatus react globally by changing its shape between flat and curved geometries? Do conditions exist where the trans and cis faces react differently? If PI4P loss is the main effect promoted by short chain ceramide treatment and mostly affects clathrin adaptors recruitment, one would expect that the trans Golgi would be more affected than the cis Golgi.

7) If the loss of coats is indeed at roots of the effect observed here, what does happen when BFA or other inhibitors of GEFs are added such as to prevent not only clathrin but also COPI from assembling at the rims? Along the same line: what does happen when a PI4P kinase inhibitor is added (e.g. PIK93)?

8) The use of the segregation between p230 and TGN46 as an index of the curled geometry seems quite indirect. What is the relationship between this segregation and the cisternae morphology?

9) Does the treatment of cells with ceramide alter the membrane bending rigidity of the Golgi or just promote the release of clathrin? In this respect, I found the Abstract and title quite misleading. The authors put a lot of emphasis there into nanodomain formation and lipid biosynthesis. Therefore, it is tempting for the reader to think that a main factor at play here would be domain formation. Instead, the authors show that the surface density of lipid domains and their size have only minor effects on the cisternae <> curled transition and that the most important factor is the coat coverage. Therefore, the tone of the Abstract should be changed. There is no indication that lipid domains play a role here.

---

## [Author Response]

*Reviewer #1:*

*1) The main assumption of the model is that "The equilibrium configuration of a cisterna is assumed to correspond to the free energy minimum." The author should explain the extent to which free energy minimisation is meaningful for such non-equilibrium structure as the Golgi apparatus, which is characterised by fluxes of proteins and lipids.*

We agree that the Golgi apparatus is a highly dynamic organelle, which is continuously subject to inward and outward membrane fluxes (e.g. by means of fusion/fission of transport carriers). Remarkably, despite the extensive exchange of membranes, the characteristic stacked-cisternae morphology of the Golgi complex is maintained during interphase.

The major assumption of our model is that the time scale of mechanical equilibration of the cisterna shape is much shorter than that of the changes of the membrane lipid and protein composition through the fluxes. Therefore, the cisternae shape is assumed to be mechanically equilibrated for every instant composition. Since the composition is in a steady state, the shape is in mechanical equilibrium corresponding to this steady state composition. Non-equilibrium shapes should be considered only if the composition would change faster than the shape relaxed to the new equilibrium state.

We estimated, on one hand, the mechanical relaxation time as a combination of the characteristic viscosity, bending rigidity, and length scale of the cisterna, τmech=ηR3/κ ~ 1 ms (see Allain et al. Phy. Rev. Lett. 2014). On the other hand, the rates of the composition changes based on the fluxes through the Golgi cisternae have been theoretically inferred from experimental data to be of the order of k ~ 0.2−0.3 min−1 (Dmitrieff et al. PNAS, 2014). Hence, a characteristic compositional relaxation time by means of membrane fluxes would be τflux ~ 100 s. Since τmech≪τflux, our assumption is therefore justified. However, it should be mentioned that the rate of bulk ER export has been estimated to be around thousand carriers per second (Wieland et al. Cell, 1987), and therefore fusion of incoming carriers to the Golgi complex could potentially be associated with a similar time scale as that of mechanical relaxation of the Golgi morphology (discussed in Sens and Rao, Methods in Cell Biol., 2013). Nevertheless, our treatment with short-chain ceramide causes the inhibition of protein export from the Golgi membranes to other intracellular destinations (Duran et al. EMBO J, 2012). Inhibition of Golgi-to-ER recycling would eventually lead to slowed-down ER export, thus decreasing the flow of membranes arriving at the Golgi with a concomitant increase in the characteristic compositional relaxation time.

In the revised version of our manuscript we now discuss this assumption in more detail (subsection “Prediction of flat and curled cisterna configurations”).

*2) The modelling strategy and results, in particular the existence of bistability, are not particularly new for membrane systems. Nevertheless, its application to Golgi cisternae shapes is to my knowledge new. The model and the derivation of the results are well explained. I find it however very unfortunate that the results are mostly presented as numerical results, varying a handful of the many model parameters. It would be much more attractive to have analytical expression for the boundaries of the phase diagram. The full model can obviously not be analytically solved, but the main concepts can be included in a much simpler, and solvable model. The essence of the model is a bendable circular domain with edges. This is essentially the same situation as a membrane domain with a line tension, as is discussed by R. Lipowsky (J. Phys. 1992, 2,1825) and many others. This model also predicts bistability (Figure 3 in Lipowsky's paper) where flat and invaginated domains can coexist. This reference should be cited and discussed.*

We appreciate the suggestion of the reviewer to include analytical approximations of the numerical results of our model. We included in the revised version of our manuscript an Appendix with analytical approximations of the model, which allowed us to estimate the flat-to-curled and curled-to-flat transition curves.

We fully agree that the nature of our model (in terms of bending energy) can be summarized as a tug-of-war between the bending energy of the central part of the cisterna and the bending energy of the rim, the latter being effectively interpreted as an edge energy. Lipowsky applied this idea to study the budding of membrane domains associated with a line tension at their interface with the surrounding membrane, and his model predicted regions of shape bistability between flat or slowly curved membranes and buds (Lipowsky, J. Phys, 1992). In our model, the "line energy" (corresponding to the bending energy of the cisterna rim) can be characterized by a line tension, which is not constant but depends on the shape of the cisterna (see e.g. Equation (A7) in the Appendix), thus preventing full closure of a highly curled cisterna due to the large negative curvature (and therefore large bending energy penalty) of the Golgi rim for very small values of rgap. This and other related papers and the relationship between them and our model have been now properly discussed in the text (subsection “Prediction of flat and curled cisterna configuration”, fourth paragraph; Appendix subsection “Approximate analytic expression for the total bending energy of a cistern”).

*3) The physical mechanism underlying this paper is not new, but its application to the shape of closed membranes (vesicles, cisternae) is new, to my knowledge. There is however a vast literature on the shape of vesicles, including stomatocyte shape not dissimilar to the curled cisternae, but driven by completely different mechanisms (such as area to volume ratio variation, see for instance U. Seifert Adv Phys. 1997, 46, 1 – Figure 14). This is not discussed at all in this paper, which is rather unfortunate. This paper should make clear that it proposes one possible scenario to explain the observations, but that other scenarii exists.*

We apologize for not discussing properly the papers mentioned by the reviewer in our manuscript. In the revised version, they are extensively discussed in relation to our findings (subsection “Qualitative description of the proposed mechanisms of SM-regulated Golgi morphology”, first paragraph).

Actually, as reviewer #3 mentions, we previously thought that a reduction in the volume-to-surface ratio of the Golgi cisternae could mechanistically explain our observations (Duran et al. EMBO J, 2012). However, our observations indicate no obvious change in the volume-to-surface ratio during the flat-to-curled Golgi cisternae transition (van Galen et al. EMBO J, 2014 and this study) and compelled us to reevaluate our previous hypothesis. Moreover, typical Golgi cisternae have a very low volume-to-surface ratio. We can estimate the dimensionless reduced volume (as defined in Seifert's review) for standard geometric characteristics of a Golgi cisterna to be in the order of v≈0.002, whereas typically the transition from oblate shapes (akin to our flat Golgi cisterna) to stomatocyte shapes (akin to our curled Golgi cisterna) occurs at much larger volume-to-area ratios, v≈0.6. We have therefore argued that the characteristic flat shape of a Golgi cisterna likely requires rim-stabilizing agents.

*4) Furthermore, previous work describing energetic description of Golgi cisternae, including phase separation of membrane components (Derganc et al. Traffic 2006; 7: 85-96, Derganc Phys. Biol. 4 (2007) 317-324), should definitely be cited and the present results compared to the existing one when applicable, in particular concerning the curvature-coupled lateral segregation.*

We thank the reviewer for bringing our attention to these papers. In our revised manuscript we cite, discuss and compare these papers in relation to our results (subsection “Evaluation of the effect of diacylglycerol on the cisterna shape transition”, last paragraph; subsection “(ii) Physical description of the free energy of a Golgi cisterna”, second paragraph; Appendix subsection “Analytical considerations of the effect of multiple stacked cisternae on the shape diagram”, first paragraph).

In brief, in Derganc et al. Traffic 2006, the authors presented a model to understand the physical mechanisms setting the number of cisternae per Golgi stack, where they compared the cisternae bending energy with the adhesion energy keeping the stack together. In Derganc, Phys. Biol (2007), the author presented a model to analyse whether membrane curvature can promote lateral partitioning of membrane constituents. The author showed that, for a flat cisterna configuration, partitioning of curved lipids is, in general, relatively small (of the order of 10%), in agreement with our computations on curvature-based DAG repartitioning. Moreover, the author also showed that stiff membrane constituents hardly partition away to the flat region of the cisterna, which is in line with our results. In these studies, the authors did not considered changes in the overall shape of a cisterna, which is one of the new contributions of our model.

*5) The theory part of the Materials and methods section gives a full geometrical description of the cisterna morphology, yielding rather cumbersome expressions (for ex. Equation 5). While these expressions are probably useful, approximate expressions allowing to derive the transition boundaries in Figure 2 and others would be more useful. A large part of the paper is devoted to explaining the effect of varying some parameters while keeping other fixed. This a rather arbitrary numerical exercise. For instance, the phase diagram of Figure 2 shows the effect of the spontaneous curvature (1/nm), while this parameter only appears in the energy coupled to one or several length scales (certainly the rim radius, and possibly the size of the flat region). Working out analytical expression with some simplifying assumptions would provide much more insights. Using the approach of a circular domain with line tension (dependent on the bending rigidity and the geometry of the rim), and neglecting the entropy of the nanodomains, it is quite easy to find analytical expression for the loss of stability of the flat cisterna. This shows that if the bending rigidity of the rim and the flat part are the same, this parameter drops out and the threshold spontaneous curvature times the rim radius has a simple expression in terms of the ratio of rim radius to cisternae radius. Such analysis should be performed and compared to the numerical results.*

We thank the reviewer for this suggestion. We now include a new Appendix in our revised manuscript with an analytical approximation for our numerical results. In particular, we now present an analytical approximation describing the cisterna shape transitions, which we compare with our numerical results (see – Figure 1). It is noteworthy that, under the simplifications discussed in the Appendix, our analytical approximation describes qualitatively well the shape transition, with a relative error of about 20%, as compared to the numerical results presented in the main text (e.g. Figure 2).

*6) Such a simplified analytical model would allow to tease apart the different contributions of the overall free energy. It is reported (subsection “Prediction of flat and curled cisterna configurations”, sixth paragraph) that the fraction of rigid domains has little impact on the transition. This observation can mean two things: either the additional bending rigidity associated to the nanodomains is irrelevant, or it is compensated by the entropy cost of partitioning. In the absence of partitioning, one can easily understand that the bending rigidity plays no role in the transition, since it is the only energy scale in the problem, as discussed above. The effect of partitioning should be assessed by comparing entropic and elastic terms analytically (even if that means using approximate expressions).*

Following the reviewer suggestions, we have made some assumptions to reach a relatively simple, approximate analytical expression for the relative contribution of the entropic and bending energies to the total free energy of the system. This analysis, as explained in the Appendix, shows that it is possible to define a critical nanodomain size, Rd*, below which entropy dominates (and very low partitioning is expected), and above which the bending energy dominates (and relatively large partitioning of the rigid nanodomains away from the Golgi rims is expected). As shown in Equation (A22) of the revised version of the manuscript, Rd* scales linearly with the rim radius. These results show that, for characteristic values of the rim spontaneous curvature and of the area fraction covered by nanodomains, the critical nanodomain size exceeds well the assumed value of 5 nm. This indicates that indeed the partitioning energy dominates over the bending energy and therefore a minor redistribution of nanodomains is to be expected, and therefore a relatively minor impact of the amount of rigid nanodomains on the shape transitions (see – Figure 2).

*7) There is a number of simplifying assumptions in the model that do not appear justified and might well influence the results.*

*7A) The most serious is, to my mind the fact that the radius of the rim is not optimised, but imposed arbitrarily. I can accept the fact that the luminal space h can be fixed by spacer proteins, the rim radius is probably fixed by some minimisation of elastic energy, and should highly depend upon the rim spontaneous curvature. Do the authors have argument to validate their assumption of rim radius independent of the rim spontaneous curvature? I would be looking for theoretical arguments, not experimental argument based on ultrastructural data. Since an energetic description is likely to suggest that the rim radius should depend on its spontaneous curvature, if it is observed experimentally that it doesn't, this would mean that some energetic contributions are not included in the current energy, thus casting doubt on the consistency of the theory with the data.*

We thank the reviewer for bringing up this point, which was not explained in the text. We agree that if the size of the rim radius would not be somehow constrained, the natural tendency of the system as a response of a decrease in the rim spontaneous curvature would be to increase its radius to match the actual spontaneous curvature. In our model, the radius of the rim cross-section is set by the distance between a protein scaffold forming the flat part of a cisterna and the cisterna edge. For example, if such scaffold would extend beyond the cisterna, e.g. if the cisterna was formed by flattening of a big liposome between two infinite flat rigid plates, the edge cross-sectional radius would be simply equal to the cisterna half-thickness, independently of the spontaneous curvature of the rim. The fact that the rim is somewhat swollen as compared to the cisterna thickness reflects the distance to which the scaffold edge approached the cisterna edge. The rim would indeed tend to swell as much as possible within this constraint to match the spontaneous curvature. In this model, the membrane spontaneous curvature could influence the detailed shape of the edge cross-section profile resulting in its deviation from the circular shape. This would be the case for the shapes where rrim is comparable to rgap. Taking into account this effect would result in corrections of the rim energy but on a semi-quantitative level of description we neglect these corrections. In our case, based on the ultrastructural data, the maximum radius of the rim allowed by the protein scaffold is rrim=30 nm, so it would stay constant as long as the spontaneous curvature is not larger than 1/rrim=0.033 nm−1, which is the case for the range of spontaneous curvatures we took into account, 0<Js,rim<0.033 nm−1, thus justifying the constant value of the rim radius throughout our calculations.

The molecular identity of such protein scaffold could be the spacer proteins maintaining the luminal thickness and/or stacking factors keeping the subsequent cisternae stacked. Interestingly, it has been experimentally shown that knockdown of the stacking/tethering factors GRASP55/65 or Golgin45/GM130 in HeLa cells leads to the cisternae unstacking, and swelling of the lumen and rims of the Golgi cisternae (Lee et al., 2014).

We explained and justified this model assumption in the revised manuscript (subsection “Prediction of flat and curled cisterna configurations”, second paragraph; subsection “(i) Geometrical description of a Golgi cisterna”).

*7B) We are dealing here with a model of a mixed membrane containing at least three different types of molecule: some "neutral" lipids, rigid nanodomains, and curvature generators. It is assumed that such mixed, three component membrane can still be described by one bending rigidity one gaussian rigidity, and one spontaneous curvature. The expression of the composite bending rigidity (Equation 12) corresponds to heterogeneous curvature distribution (i.e., the stiffer regions are less curved than the softer ones, for a given mean curvature). It is thus very likely that the total curvature energy of this mixed membrane with curvature generators will be very different from the one proposed here if the curvature generators bind preferentially to one of the two components. Apparently, SM regulates both membrane stiffness and the binding of curvature generators, so spontaneous curvature and bending rigidity are probably spatially correlated. The authors should discuss this point.*

We agree with the reviewer that if curvature generators were to be localized specifically to, say, SM-rich rigid nanodomains, the results could significantly differ from those presented in our paper. However, there are experimental arguments to consider that this might not be the case. As we mentioned in the discussion, our collaborator Giovanni d'Angelo (IBP-CNR in Naples, Italy), has recently described the metabolic pathway by which PI(4)P-binding proteins, such as the clathrin adaptor γ-adaptin, are released from the Golgi membranes upon addition of short-chain ceramide (Capasso et al. bioRxiv, 2016). This release is indirectly induced by sphingomyelin (SM) *metabolism*, since SMS-mediated conversion of ceramide to SM parallels the conversion of phosphatidylcholine to diacylglycerol (DAG). DAG recruits and activates protein kinase D (PKD), which in turn regulates the levels of PI(4)P on the cytosolic monolayer of the Golgi membranes and therefore the recruitment of PI(4)P-binding proteins, such as the clathrin adaptor AP-1. Phosphoinositides are mostly localized away from SM-rich rigid nanodomains (see e.g. Arumugan and Bassereau, Essays Biochem, 2015), thus suggesting that this signaling event is spatially uncoupled from SM-rich rigid nanodomains. Moreover, SMS-mediated synthesis of SM is restricted to the luminal side of the trans-Golgi membrane (Huitema et al. EMBO J, 2004), whereas recruitment of curvature generators occurs at the cytosolic side of the membrane. How luminal SM connects with the organization of the cytosolic leaflet of the Golgi membrane and the events occurring on that side is, to our knowledge, not well understood. Overall, with all the published data at hand we think it is reasonable to assume that there is no preferential binding of curvature generators to specialized membrane regions.

In the revised version of the manuscript we now stress these points both in the Discussion (subsection “DAG acts as a signaling effector rather than as a molecular shaper to control Golgi membrane morphology”) and when we present the model and describe the localization of membrane curvature generators (subsection “(ii) Physical description of the free energy of a Golgi cisterna”, third paragraph). In addition, we also explicitly mention that the expression of the composite bending rigidity (Equation 12) has the meaning of an average over the rigid and soft patches (V.S. Markin, Biophys. J., 1981).

*8) Related to 7B): The Abstract claims to investigate the dual effect of SM, which is to affect the rigidity of membranes by forming stiff domains, and to regulates the association of curvature-generating proteins. It is not clear to me what whether SM promotes or inhibit the recruitment of curvature generators. Is this known/implemented in the model?*

*In the model, the former is associated to a fraction Φ_budding_ and the latter to a fraction Φ, but no connection is made between these two fractions. If SM mediate both effects, should Φ_budding_ and Φ be connected?*

This would indeed be the case if SM directly recruited curvature generating proteins to the Golgi membranes, however, as mentioned above (see our answer to point 7B) this is not the case. It is actually SM metabolism (through a signaling cascade triggered by the metabolic conversion of ceramide to SM), which controls the levels of curvature generators at the Golgi membranes. We have modified the Abstract to better explain this point.

*Why is the partitioning of the nano-domain partition between the rim and the mid-section subjected to a thermodynamic treatment (enthalpy vs. entropy), but not the curvature agents? This seems a rather arbitrary choice.*

Although we agree that a thermodynamic treatment could be implemented also for the distribution of curvature generators along the membrane, we have to take into consideration that these are cytosolic proteins, which are recruited to the membrane by specific lipids and/or adaptor proteins. Moreover, additional considerations, such as the well-known curvature-sensing properties of various curvature-generating proteins (see e.g. Antonny, Annu Rev Biochem, 2011), regulate their binding affinities and subsequently affect the lateral partitioning of these proteins between the highly curved rim and the flat central part of the cisterna. Hence, for the sake of simplicity we have considered only the two extreme situations: (i) exclusive localization of curvature generators to the cisterna rim; and (ii) a homogeneous distribution of these proteins along the entire membrane of the cisternae (see Figure 2—figure supplement 1). We explained and justify our choice in the revised version of our manuscript (subsection “(ii) Physical description of the free energy of a Golgi cisterna”, fourth paragraph).

*9) When the entire cisterna including the mid part, has spontaneous curvature, one observes "only a general shift of the shape transitions between the flat and curled cisterna configurations towards the lower values of the membrane spontaneous curvature". It is not clear to me why this shift occurs, since if both sides of the cisterna have the same spontaneous curvature, its contribution to the global energy can only depend on the area difference between the convex and concave sides of the cisterna (which have curvatures of opposite sign). I would then expect the contribution of the spontaneous curvature of the mid part to the energy difference between flat and curled to be or order kappa J_s_ h~ kappa/2. This seems quite small compared to the overall energy scale, and does not explain the rather large shift observe Figure 1—figure supplement 2. Could the author provide an estimate for the effect of the spontaneous curvature of the flat part.*

See our global answer to points 9-11 below.

*10) Related to this, the shape parametrisation is clear for flat cisternae, but a little less clear for curled one (Figure 1). My understanding is that since the total are is conserved, and the area of the rim decreases upon curling, the area of the mid part must increase for curled cisternae. This increase is distributed in some way between the concave and convex sides of the cisterna. I would not expect these details to matter much, but since there is an effect of the spontaneous curvature of the mid part, that I can only understand based on the area difference between the convex and concave regions, the details might matter. Judging from Figure 1, both mid regions are defined by the same opening angle, and are hence characterised by a relative area difference Δ S/S ~ h/R. There is no real reason for this to be the case, and there is more leeway here for energy minimisation depending of the spontaneous curvature. Could the authors comment on this? If the conclusion is that these details do matter, it speaks against the general validity of the proposed energetic description.*

See our global answer to points 9-11 below.

*11) The effect above is discussed: "the results are highly insensitive to the amount of curvature generators present in the central part of the Golgi cisternae, basically due to the opposite curvature of the two parallel membranes composing this region." However, Figure 2—figure supplement 1 shows a shift of about 50% of the transition boundaries. Since this paper is a practical application of well-known concepts, rather than the description of new concepts, I guess numerical values are important and the author should provide an estimate of this shift using simple analytical models.*

Comments 9, 10 and 11 are all related and our response to all follows. These comments point to a very fair concern, and we apologize for not explaining it in more detail in our initial version of the manuscript. Indeed, we initially thought that the effect of a non-negative spontaneous curvature in the central part of the cisterna would have a negligible effect given that the changes in the bending energy of the positively curved membrane would be counter-balanced by the changes in the bending energy of the apposed, negatively curved membrane (up to a correction of the order *h/R* as the reviewer pointed out). However, as the reviewer also commented, given that the total surface area of the cisterna is constant during the deformation, cisterna curling leads to an increase in the surface area of the central part of the cisterna, concomitant with a decrease in the surface area of the rim. A non-zero spontaneous curvature of the central part of the cisterna therefore increases the total bending energy of this cisterna part, thereby penalizing cisterna curling and eventually shifting the flat-to-curled cisterna transition towards smaller values of the spontaneous curvature, as shown by our numerical results (Figure 2—figure supplement 1). In the Appendix of our revised manuscript, we present an analytical estimation of the extent of this shift, under certain approximations. Using the same parameters we used to generate Figure 2—figure supplement 1, we estimated that the spontaneous curvature of the shape boundary decreases by about 30% when the curvature generators are not forced to be solely localized at the cisterna rim, which is in good agreement with our numerical results (see Appendix). Moreover, in the revised version of the text, we clarify these findings both in the Results (subsection “Prediction of flat and curled cisterna configurations”, seventh paragraph) and model sections (subsection “(ii) Physical description of the free energy of a Golgi cisterna”, fourth paragraph).

*12) The analysis of drug treatment and recovery is the light of the bistability discussed in this paper is quite interesting. I however do not fully understand the implication of the loss, and the recovery, of TGN46 and p230 colocalisation. These proteins appear to be TGN proteins, while this paper discusses the morphology of Golgi cisternae. Furthermore, I thought that clathrin was also mostly localised at the TGN and was responsible for transport between the TGN and the endosomes/plasma membrane. Is the curling of the Golgi cisternae accompanied with structural changes of the TGN? Should we understand the Golgi and the TGN as a single structure? The concentric stacked onion-like structure presumably involves curling of all Golgi cisternae. Where is the TGN located? In the convex or concave region of the curled stack?*

We showed in Figure 6 that curling of the TGN (as monitored by an anti-p230 antibody) parallels curling of the cis-medial Golgi (observed using the MannII-GFP marker protein).

In the revised version, we have enlarged and improved the contrast of the immuno-electron microscopy images in Figure 6. The TGN is located in the concave region of the curled stack.

We previously showed that short-chain ceramide treatment had several concomitant effects in the organization and shape of the membranes of the Golgi complex and the TGN (van Galen et al. JCB, 2014). We observed by immuno-electron microscopy that the localization of the late Golgi enzyme (trans-Golgi cisternae and TGN) sialyltransferase (ST) tagged with GFP localized to the inner, concave part of the stacked onion-like structure (see e.g. Figure 2 in that reference). The second effect we observed was that such a morphological change (from flat to curled stacked cisternae) paralleled the lateral segregation of different Golgi markers, such as the cis-/medial-Golgi markers Mannosidase II and GRASP65, or the trans-Golgi membrane/TGN markers ST, p230 and TGN46, as monitored by fluorescence microscopy (see Figure 1 in that reference). In summary, our data indicated that the curling of the Golgi cisternae occurs somewhat concomitant with the segregation of different Golgi/TGN markers. However, as we explain in more detail in response to comment 14, the exact relation between protein segregation and morphology changes still remains unresolved. We have now expanded this point in the main text (subsection “Experimental evidence for hysteresis of Golgi cisternae morphology during recovery from short-chain ceramide treatment”, fourth paragraph) to facilitate understanding of our experimental approach.

In addition, we would like to emphasize that we do not claim that the entire Golgi complex/TGN morphological transition is due solely to the loss of clathrin, which, as the reviewer pointed out, mainly localizes to the TGN to form export carriers. Whether there are other curvature-inducing proteins involved in this process is still unknown, as we already mentioned in the Discussion in our original version. We now addressed this part in more detail to clarify our message (subsection “Membrane curvature generators dynamically stabilize the flat shape of Golgi cisternae”, first paragraph).

*13) Along the same lines: the curling phenomenon is a spontaneous symmetry breaking, which predict an equal likelihood to curl one way or the other. Is curling equally likely in both direction (i.e. toward or away from the TGN) in experiments? If there is a net tendency to curl one way, that means that symmetry is broken from the start by something. This could be because the line tension effect mostly impact the TGN, which then drive the entire Golgi to curl, but in this case, the energy function should include the other Golgi cisternae. It could also be that symmetry is broken by anterograde or retrograde fluxes, in which case the entire model is problematic. More discussion on this is mandatory.*

Our electron microscopy analysis shows that curling occurs towards the TGN as we explained in point 12 above. Interestingly, a previous study (Martinez-Alonso et al., Traffic, 2005) showed that a morphologically similar curling of the Golgi/TGN stack occurs when HeLa cells are incubated at 15ºC, and that this shape transition is accompanied by the release of COPI proteins from the *cis*-Golgi cisternae. In that case, the curling direction was the opposite, towards the *cis*-side of the stack. The reasons for this symmetry breaking can be manifold, as the reviewer already indicated, but given our experimental evidence we favor a situation where this is driven by an initial release of rim stabilizers from the trans-side of the stack.

Given our experimental observations, and following the suggestion of the reviewer, we have now expanded our analysis to account for the various cisternae of the Golgi complex. Specifically, we propose a system composed of *N* cisternae, each of them characterized by a free energy of bending and partitioning. The different cisternae are connected along the stack by certain stacking factors, which we assume (based on our experimental observations) are strong enough to impose that all cisternae curl equally. For the sake of this argument, we disregard the effects of nanodomain partitioning (that is, we assume there is no nanodomain partitioning Φrim=Φmid=Φ). In addition, we need to ascribe the distribution of the spontaneous curvature along the different cisternae of the stack, and how this distribution changes as a result of the short-chain ceramide treatment. Unfortunately, we have no direct experimental data on this, and despite the fact that we observed the release of clathrin from the trans-Golgi membranes upon short-chain ceramide treatment, we do not currently know whether other curvature generators/stabilizers are being released from the early Golgi membranes or not. Taking into account this limitation, we considered two situations in our analysis. First, we assume that the spontaneous curvature of the rim is equal for all Golgi cisternae, and that short-chain ceramide treatment leads to a similar reduction in the spontaneous curvature of all the cisternae. Possibly, the dynamics of the release of the different bending effectors is slightly different, being faster for the TGN membranes thus initiating the curling process and setting the curling direction, which is later followed by the rest of the cisternae. Under these assumptions, the free energy of the whole stack is simply the free energy of a single cisterna multiplied by the number of cisternae, and the whole analysis is independent of the number of cisternae, thus justifying our analysis considering a single isolated cisterna. The second situation that we consider corresponds to a model consisting of t*trans*-like cisternae with a characteristic rim spontaneous curvature, Js,rimtrans, and N−t
*cis/medial*-like cisternae, characterized by a different rim spontaneous curvature, Js,rimcis. Using an approximate analytical model, we then calculate the shape transition boundaries as a function of the two spontaneous curvatures. These results, as expected, show that when the two curvatures are equal, the shape transitions are independent on the number of cisternae and correspond to the values obtained for a single cisterna (see Appendix, and – Figure 3).

We now include a more comprehensive explanation on how we propose that the direction of curling is initiated (subsection “Short-chain ceramide treatment causes the release of clathrin coats from the Golgi membranes prior to cisterna curling”, last paragraph and subsection “Membrane curvature generators dynamically stabilize the flat shape of Golgi cisternae”, first paragraph) as well as the analysis of the multiple cisternae scenario in the Appendix of the revised manuscript.

*14) In the mind of the authors, is the lateral segregation discussed upon D-cer-C6 treatment related to a lateral partitioning such as the one quantified in Figure 3? In any case, the observation that colocalisation is back to untreated level after 16h in a 30' treatment (Figure 6), while about half the Golgi stacks still show a curled shape (Figure 6) seems to suggest that partitioning and curling are not directly related, unlike the author claim.*

The mechanisms that leads to protein segregation upon short-chain ceramide treatment still remain unknown. We previously showed that such segregation (e.g. between the Golgi enzyme sialyltransferase and its substrate TGN46) has functional consequences, such as defects in secretory cargo glycosylation (van Galen et al. JCB 2014). We hypothesized that changes in the physico-chemical properties of the membranes induced by the treatment (such as changes in the bilayer thickness) could induce the segregation of such proteins, but our preliminary unpublished data suggests this is not the case. Alternatively, decreased nanodomain levels and increasing partitioning could be a factor driving this segregation, as we proposed in our discussion. However, to be fair, this still remains an open question, which we intend to address in the future.

In relation to the second part of the reviewer comment, we fully agree that the correlation between lateral segregation of Golgi markers and cisternae morphology is not 100%. We think that this occurs because during D-cer-C6 washout three different processes occur simultaneously albeit with different dynamics (see Figure 7): (i) recovery of normal clathrin levels at the Golgi membranes (which, according to Figure 6, takes about 2h for a 30' ceramide treatment); (ii) recovery of the amounts of nanodomains (which, by assuming a direct correlation with the recovery of the Golgi marker localization, takes about 12 hours, according to Figure 6); and (iii) recovery of the flat Golgi morphology (which is a stochastic process that requires crossing an energy barrier as we discuss in the main text, and which takes at least 16 h, according to Figure 6). To clarify these points, we now explain this hypothesis in detail in the revised manuscript (subsection “Experimental evidence for hysteresis of Golgi cisternae morphology during recovery from short-chain ceramide treatment” and Discussion, third paragraph).

*15) The level of clathrin in the Golgi reaches its lowest level after about 30'. This suggest that the loss of the curvature generators saturates after that exposure time (Figure 6). However, the recovery of colocalisation is much faster (12h) after 30' treatment than after 4h treatment (12h vs. 24h). This suggest that the recovery time depends on the treatment duration, while the bistability picture would suggest that it only depends on the energy barrier, which is related to the clathrin level, and should thus saturate after 30'. Can the author comment on this, which in my mind argues against the bistability picture?*

We understand the concerns of the reviewer regarding this apparent contradiction. Indeed, as the reviewer correctly stated before, the level of Golgi protein segregation does not perfectly correlate with the curling of the Golgi cisternae. In the case of protein co-localization, the recovery, as we proposed, would depend on the recovery of the levels and distribution of SM-rich nanodomains, which we expect would indeed depend on the duration of the short-chain ceramide treatment.

In respect to Golgi morphology, as the reviewer comments, our results indicate that it mainly depends on the levels of curvature generating proteins at the membrane, and therefore, once the system has returned to the initial levels of clathrin on the membrane after short-chain ceramide washout, the recovery time for Golgi flattening should not depend on how long the treatment was. However, the total time required for the recovery of the flat morphology during the washout is the sum of the time it takes to overcome the energy barrier plus the time it takes to recover the initial clathrin levels at the Golgi membranes. To a first approximation, the former time, is indeed independent of the duration of the ceramide treatment. However, the latter time could depend on the signaling events leading to the recovery of PI(4)P and hence clathrin levels. To test this, we treated cells for 4h with D-cer-C6 treatment, and monitored the recruitment of clathrin to the Golgi membranes during the washout. We observed that this recovery time is longer (about 6h) than for a 30' treatment, therefore predicting an extra time to recover Golgi morphology in this situation (new Figure 6—figure supplement 1).

In addition, one also has to take into account that the characteristic time for stochastic transition, according to Arrhenius kinetics, grows as eΔF/kBT. Thus, using the estimated parameters, a change in the recovery time from 12 h to 24 h only corresponds to a 4% increase in the energy barrier (from 17.6 to 18.3 k_B_T). Such a subtle difference can be ascribed to different factors that might depend or change during the prolonged treatment with short-chain ceramide, such as small changes in the cisterna area.

We included a new supplementary figure and discussion on this in the revised manuscript (subsection “Experimental evidence for hysteresis of Golgi cisternae morphology during recovery from short-chain ceramide treatment”, second paragraph; Discussion, third paragraph and subsection “Membrane curvature generators dynamically stabilize the flat shape of Golgi cisternae”, second paragraph).

*Reviewer #2:*

*Theoretical part*

*1) The authors write: 'The equilibrium configuration of a cisterna is assumed to correspond to the free energy minimum'. Given the emphasis put in local energy minima later in the text, this statement is a little bit confusing.*

We changed this sentence in the revised version of the manuscript, also following the suggestion from reviewer 2 (see our response to comment 1), to be more precise in our wording.

*2) The authors write: 'the results, presented in Figure 2—figure supplement 2, show that both the energy barrier required to flatten a curled cisterna, Δ𝐹𝑐𝑢𝑟𝑙−𝑓𝑙𝑎𝑡, and the energy barrier required to curl a flat cisterna, Δ𝐹𝑓𝑙𝑎𝑡−𝑐𝑢𝑟𝑙, decrease with the amount of rigid nanodomains on the Golgi membranes. It seems to me that what is shown here is rather an increase.*

We apologize for the confusion. Indeed, we mean that the energy barriers decrease when the amount of nanodomains also decreases. We have changed the wording to clarify better the text.

*3) In a previous study, the authors proposed an alternative mechanism for the flat to curled shift: “We suggest that these changes are a consequence of the effect of d‐ceramide‐C6 on membrane fission versus membrane fusion at the Golgi membranes. In the presence of d‐ceramide‐C6, transport carriers fuse with the Golgi membranes, however, membrane export is blocked. This increases the surface area‐to‐volume ratio of the Golgi cisternae. This is similar to morphological changes in model membranes as a result of an increase in the surface area‐to‐volume ratio.” Why is this hypothesis no longer considered here?*

We indeed previously thought that an increase in the surface-to-volume ratio of the Golgi cisternae could mechanistically explain our observations (Duran et al. EMBO J, 2012). However, our later observations indicated no obvious change in the volume-to-surface ratio during the flat-to-curled Golgi cisternae transition (van Galen et al. EMBO J, 2014 and this study), which led us to propose new hypotheses to describe cisterna curling. Moreover, typical Golgi cisternae have a very low volume-to-surface ratio, which according to previous theoretical models cannot explain flat cisterna shapes, but likely requires rim-stabilizing agents (see also our response to comment 3 from reviewer 2). We discuss this point now in detail in the text (subsection “Qualitative description of the proposed mechanisms of SM-regulated Golgi morphology”, first paragraph and subsection “Short-chain ceramide treatment causes the release of clathrin coats from the Golgi membranes prior to cisterna curling”, last paragraph).

*4) A few theoretical papers on the discoïdal shape of membranes or Golgi stacks exist, notably by Derganc, Lipowski, and Svetana. Please comment, compare.*

We thank the reviewer for bringing these papers to our attention. This comment has been brought up also by reviewer 2 (see also our responses 2 and 4 to his comments). We now discuss them in the revised manuscript. Moreover, we now included an Appendix with an analytical approximation of our model, which allowed us to compare our model with the previous work the reviewer mentions.

*5) In terms of curvature, is the curled shape so highly curved as stated in the Abstract? (…affecting SM homeostasis converted flat cisternae into highly curved membranes). Looking at the EM, the curvature of the cup shape cisternae seems rather modest.*

We meant that the cup-like shape of the cisternae was highly "curled" or highly curved relative to the initial flat configuration. Indeed, this curvature is still relatively small as compared to other high curvatures in cells (Golgi rims, transport vesicles, etc.). Hence, we changed "highly curved" for "highly curled" in the revised Abstract.

*For the experimental part I have the following concerns:*

*6) The lipid composition and the nature of protein coats associated with the cis and trans faces of the Golgi apparatus are very different. Therefore, why does the Golgi apparatus react globally by changing its shape between flat and curved geometries? Do conditions exist where the trans and cis faces react differently? If PI4P loss is the main effect promoted by short chain ceramide treatment and mostly affects clathrin adaptors recruitment, one would expect that the trans Golgi would be more affected than the cis Golgi.*

We thank the reviewer for this comment, which has been also raised by reviewer 2 (see also our response to his comments 12 and 13), and it was not sufficiently clear in our original submission.

We would like to emphasize that we do not claim that the entire Golgi complex/TGN morphological transition is due solely to the loss of clathrin from the TGN membranes. Indeed we now present in the Appendix new calculations on a multiple cisternae model, which indicate that stack curling indeed requires the loss of curvature effectors from multiple, if not all, cisternae (see – Figure 3). Whether there are other curvature-inducing proteins involved in this process is still unknown, as we mentioned in the Discussion of our original version. We now discussed this in more detail in the revised manuscript (subsection “Membrane curvature generators dynamically stabilize the flat shape of Golgi cisternae”, first paragraph).

*7) If the loss of coats is indeed at roots of the effect observed here, what does happen when BFA or other inhibitors of GEFs are added such as to prevent not only clathrin but also COPI from assembling at the rims? Along the same line: what does happen when a PI4P kinase inhibitor is added (e.g. PIK93)?*

Inhibition of COPI vesicle formation by adding BFA induces, within minutes, the tubulation of the Golgi membranes and their subsequent fusion to the ER, therefore preventing any slower dynamic processes such as Golgi curling. Importantly, we previously showed that short-chain ceramide-treated cells were basically insensitive to BFA in terms of aforementioned relocalization of Golgi membranes to the ER (Duran et al. EMBO J, 2012). However, Martinez-Menarguez and colleagues showed that when HeLa cells are incubated at 15ºC to block protein secretion at the level of the ERGIC, components of the COPI coat (e.g. β-COP) are released from the early Golgi membranes (Martinez-Alonso, Traffic, 2005). Strikingly, they observed by cryo-electron microscopy that the morphology of the Golgi complex turned into a curled shape analogous to the one we observed upon short-chain ceramide treatment. Moreover, they observed that the curling of the cisternae occurred towards the *cis* side, oppositely to what we observed, which makes sense in the view that curling starts at the cisterna most affected by the treatment away from the stack, which is then followed by the remaining cisternae (see also our response to comment 13 of reviewer 2).

Regarding the role of PI4-kinases in this process, we would like to point to the paper by D'Angelo's group (Capasso et al. BioRxiv, 2016), where the authors actually mapped the signaling cascade of the events leading to clathrin release upon short-chain ceramide treatment. Essentially they showed that by blocking PI4-kinases at the Golgi either by chemical inactivation (using PIK93) or by RNAi sensitized the cells against short-chain ceramide treatment, meaning that the levels of PI(4)P and their binding effectors on the Golgi membranes were unaltered. Moreover, it has been recently reported that knockdown of the two Golgi-localized PI(4) kinases in Atg5 knockout cells induces curling of the Golgi cisternae (Yamaguchi et al. EMBO J, 2016).

We have now expanded these points in the revised manuscript (subsection “Short-chain ceramide treatment causes the release of clathrin coats from the Golgi membranes prior to cisterna curling”, last paragraph and subsection “Membrane curvature generators dynamically stabilize the flat shape of Golgi cisternae”, first paragraph).

*8) The use of the segregation between p230 and TGN46 as an index of the curled geometry seems quite indirect. What is the relationship between this segregation and the cisternae morphology?*

We agree with the reviewer that using the level of colocalization between p230 and TGN46 is quite an indirect readout of Golgi cisternae morphology. To avoid any confusions about this point, we now clearly state that short-chain induced protein segregation at the Golgi complex, although it correlates somehow with the changes in cisternae morphology, is driven by a yet to be identified mechanism.

We stressed this in the revised version (subsection “Experimental evidence for hysteresis of Golgi cisternae morphology during recovery from short-chain ceramide treatment”, fourth paragraph and Discussion, third paragraph). Also, see our response to point 12 of reviewer 2.

*9) Does the treatment of cells with ceramide alter the membrane bending rigidity of the Golgi or just promote the release of clathrin? In this respect, I found the Abstract and title quite misleading. The authors put a lot of emphasis there into nanodomain formation and lipid biosynthesis. Therefore, it is tempting for the reader to think that a main factor at play here would be domain formation. Instead, the authors show that the surface density of lipid domains and their size have only minor effects on the cisternae <> curled transition and that the most important factor is the coat coverage. Therefore, the tone of the Abstract should be changed. There is no indication that lipid domains play a role here.*

We agree that the role of nanodomains in controlling the shape and shape transitions of the Golgi cisternae is minor. However, the numerical results of our model show that curling of a Golgi cisterna is accompanied by a larger degree of lateral segregation of nanodomains (see Figure 3). We believe that this prediction has some potentially interesting implications as we mentioned in the Discussion, which will be hopefully experimentally tested in the future. Nevertheless, following the suggestion of the reviewer we have changed the Abstract to somewhat temper the role of nanodomains.